# Analysis of Spatial Suitable Habitats of Four Subspecies of *Hippophae rhamnoides* in China Based on the MaxEnt Model

**DOI:** 10.3390/plants14111682

**Published:** 2025-05-31

**Authors:** Mengyao He, Fanyan Ma, Junjie Ding, Panxin Niu, Cunkai Luo, Mei Wang, Ping Jiang

**Affiliations:** 1College of Agriculture, Shihezi University, Shihezi 832003, China; 16699042101@163.com (M.H.); mafanyanshzu@163.com (F.M.); m13899502568@163.com (P.N.); lck444973413@163.com (C.L.); 2Xinjiang Yuli Desert Ecosystem Observation and Research Station, Yuli 841500, China; 13677568910@163.com

**Keywords:** *Hippophae rhamnoides*, climate change, MaxEnt, potential distribution

## Abstract

*Hippophae rhamnoides* L. is an ecologically and medicinally significant species widely distributed across Eurasia, the suitable habitat of *H. rhamnoides* subsp. *sinensis* (is hereinafter referred to as *sinensis*) is concentrated in Northwest and Southwest China (approximately 34–40° N, 100–115° E). *H. rhamnoides* subsp. *yunnanensis* (hereinafter referred to as *yunnanensis*) is mainly distributed in the Hengduan Mountains and surrounding areas (approximately 25–30° N, 98–103° E). *H. rhamnoides* subsp. *mongolica* (hereinafter referred to as *mongolica*) is native to Central Asia to Siberia and is mainly distributed in Northern Xinjiang and Western Inner Mongolia in China (approximately 40–50° N, 100–120° E). *H. rhamnoides* subsp. *turkestanica* (hereinafter referred to as *turkestanica*) is mainly distributed in Western Xinjiang (approximately 40–45° N, 70–85° E). Climate change poses a considerable challenge, affecting its distribution and leading to shifts in its habitat ranges. This study applies the MaxEnt model to assess climate-driven distribution patterns of *Hippophae* species in China, and predicts current and future suitable habitats under climate change scenarios. This study employs the MaxEnt model and ArcGIS to simulate the potential distribution of four subspecies of *H. rhamnoides* during the current period and future projections under scenarios SSP1–2.6 and SSP5–8.5. The analysis reveals that the distributions of *sinensis*, *mongolica*, *yunnanensis*, and *turkestanica* are influenced primarily by climate variables such as temperature and precipitation, while *yunnanensis* is predominantly restricted by altitude. Future projections indicate that under the extreme climate of SSP5–8.5, centroid migration will be disrupted; specifically, *sinensis* is expected to migrate northeast or oscillate, *mongolica* will expand southwest but be limited by desert steppe zones, and *turkestanica* may face risks associated with groundwater depletion. This study advocates for integrating climate, ecological, and genetic data into conservation planning, with an emphasis on groundwater restoration and exploring genetic resources for stress resilience. The insights offered here contribute significantly to understanding climate adaptation mechanisms in arid and mountainous ecosystems and guide biodiversity conservation efforts.

## 1. Introduction

*Hippophae rhamnoides* L., belonging to the Elaeagnaceae family and also known as sea buckthorn, is a perennial deciduous shrub or small tree with medicinal and nutritional value [1]. Phytochemical studies have revealed a variety of phytonutrients [2,3]. As an economic plant, *Hippophae rhamnoides* is both edible and medicinal, and it thrives on arid, barren land, offering advantages difficult to match by other economic plants [4]. The ecological and economic value of *Hippophae rhamnoides* substantially surpasses conventional understanding. In China, the direct economic valuation of a hectare of *Hippophae rhamnoides* plantation amounts to RMB 3100, while its ecological value reaches RMB 68,000 per hectare, accounting for 95.66% of the total value [5]. Internationally, Uganda’s *Hippophae rhamnoides* butter export venture currently produces 800,000 L annually, with the potential to scale up to 80 million liters [6]. In Poland, *Hippophae rhamnoides* is utilized as an energy crop, demonstrating a biomass calorific value of 17.5 MJ/kg and exhibiting a 30% higher energy output per unit area compared to conventional crops, thus serving as a significant supplement to renewable energy systems [7]. Sea buckthorn enhances soil stability through its “shallow dense deep extended” root system, and its canopy and litter can regulate hydrological processes to reduce erosion risk [8,9]. At the same time, the nitrogen fixation effect of root nodules further enhances soil resistance to erosion [10]. However, widespread early decline or death of *Hippophae rhamnoides* has severely impacted their ecological and economic benefits. In addition to tree aging, disease and pest damage, and poor management, there are other factors that were not adequately considered in early plantation forest designs, such as ecological geographic distribution and habitat adaptability [11]. At the same time, climate change has caused significant shifts in the global ecosystem, particularly affecting species’ geographical distribution, physiology, and population dynamics [12]. The relationship between plant habitat conditions and climate change has become a major focus in global climate change and phytogeography research [13]. To address the challenges of adaptation and species distribution shifts due to climate change, studying potential species distribution is crucial for developing biodiversity conservation strategies [14].

We generally use species distribution model (SDM) to predict changes in species adaptation to climate, based on the ecological niche principle, so it is also known as habitat suitability model or ecological niche model [15]. Among SDMs, the MaxEnt model is simple to operate and effective, based on the principle of maximum entropy in statistical mechanics, generating prediction results that are closest to a uniform distribution while satisfying environmental variable constraints such as mean and variance, and it limits model complexity through regularization, effectively preventing overfitting [16,17]. Its prediction accuracy is not significantly affected by sample size, making MaxEnt one of the most widely used and reliable SDMs globally [18,19]. Tu [20] demonstrated superior predictive performance of the MaxEnt model over other models, such as GAM (Generalized Additive Models), GBM (Gradient Boosting Machines), GLM (Generalized Linear Models), and CTA (Classification Tree Analysis), based on machine learning algorithms. AUC/TSS serve as the primary quantitative indicators, with AUC > 0.9 and TSS > 0.8 established as high-precision benchmarks [21]. During repeated runs, lower standard deviations of AUC indicate higher stability, and among the nine models evaluated, MaxEnt demonstrates superior performance with the lowest standard deviation (ranging from 0.02 to 0.05) [22].

Previous studies by Zhang et al. [11] and Wang et al. [23] laid the research foundation for *sinensis* using the MaxEnt model, while Xie et al. [24] revealed the distribution trend of *yunnanensis* under future climate change. In China, the genus *Hippophae rhamnoides* comprises four subspecies, which are *sinensis*, *mongolica*, *yunnanensis*, and *turkestanica* [25], and previous studies have not fully predicted the distribution of the entire genus. However, the research is plagued by two key limitations: 1. It independently models each subspecies while neglecting the niche overlap mechanism between *mongolica* and *turkestanica*; 2. Provincial-scale models (such as those for the *yunnanensis*) are inadequate for informing spatial decision-making in national ecological barrier planning.

This paper employs the MaxEnt model to analyze the relationship between geographic distribution and climate, visualize suitable habitat areas, and predict the potential distribution and range shifts of the entire genus in China. These findings provide a scientific basis for conservation planning and management strategies.

## 2. Results

### 2.1. Model Accuracy Assessment and Contribution of Environmental Variables

Following an initial evaluation of 19 bioclimatic variables, 10 key environmental factors were identified for model assessment. The MaxEnt model yielded an AUC value of 0.907, indicating high predictive performance and reliability. The contribution rates of environmental variables to the potential habitat distribution of *Hippophae rhamnoides* were ranked as follows (Table 1).

Plant growth and distribution are closely linked to hydrothermal conditions, which serve as fundamental determinants of plant reproduction and survival. This study identified temperature, precipitation, and altitude as the primary environmental drivers influencing the distribution of *H. rhamnoides*. Specifically, *sinensis* was predominantly influenced by bio12, bio14, and bio11. Based on the response curves of key environmental variables, the optimal habitat conditions for *sinensis* under current climatic conditions were as follows: bio11: −6.24 to 3.59 °C, bio12 ranged from 506.38 to 1052.38 mm, and bio14 ranged from 1.08 to 5.24 mm.

The dominant environmental factors affecting *mongolica* were bio13, altitude, and bio17. Specifically, the suitability ranges were bio13, ranging from 31.78 to 33.61 mm, altitude ranging from 741.91 to 2222.73 mm, and bio17, ranging from 15.75 to 20.17 mm.

The distribution of *yunnanensis* is primarily influenced by several key environmental variables. The altitude ranges from 2673.62 m to 4018.12 m, while the temperature seasonality, measured by bio4, falls between 532.92 and 642.78. Additionally, the mean temperature of bio11 varies from −2.99 °C to 5.09 °C.

The dominant environmental factors shaping the distribution of *turkestanica* were bio13, bio2, and bio6, of which bio13 was 40.05–40.84 mm, bio6 was −16.79–8.39 °C, bio2 is 12.41–14.04 °C (Table 2).

### 2.2. Current Distribution Analysis of Potentially Suitable Habitat

The predicted distribution of *H. rhamnoides* was analyzed and visualized using ArcGIS software. Distribution maps of the potential suitable habitat under two different climate scenarios, SSP1–2.6 and SSP5–8.5, were generated for the present period, the 2050s, and the 2070s (Figure 1).

The current distribution of four *Hippophae rhamnoides* subspecies which are primarily concentrated in Central and Northwestern China. A comparison of the current distribution reveals a significant discrepancy between the observed distribution of *Hippophae rhamnoides* and its predicted potential habitat, a mismatch that could be further exacerbated by future climate change (Table 3).

### 2.3. Future Potential Distribution Under Climate Change Scenarios

The spatial distribution and potential suitable habitat change in the four *Hippophae rhamnoides* subspecies are projected to change significantly under different climate scenarios. The color-coded graphs depicting the potential changes in suitable habitat area illustrate the shifts in suitable areas for each subspecies under current and future climate conditions. Under the SSP1–2.6 scenario, the projected changes in the expansion areas are as follows: *sinensis* (−3.4285 × 10^4^ km^2^), *mongolica* (−0.7725 × 10^4^ km^2^), *yunnanensis* (−0.7829 × 10^4^ km^2^), and *turkestanica* (−1.4444 × 10^4^ km^2^). Similarly, under the SSP5–8.5 scenario, the suitable habitats are projected to decrease by: *sinensis* (−2.0763 × 10^4^ km^2^), *mongolica* (−5.2152 × 10^4^ km^2^), *yunnanensis* (−3.7395 × 10^4^ km^2^), and *turkestanica* (−8.7378 × 10^4^ km^2^). These results consistently indicate a contraction of suitable habitats across all subspecies under both climate scenarios, emphasizing the potential vulnerability of these species to future climate change.

The future distribution of *sinensis* Rousi is projected to exhibit similar patterns to the current distribution, but with varying trends under different climate scenarios. Highly suitable areas are predominantly located in Central China, with medium-suitability areas surrounding them. Under the SSP1–2.6 scenario, significant changes in suitable habitat are projected. Between 2021 and 2040, and the subsequent decade from 2041 to 2060, the total area of suitable habitat is expected to decrease by 37.41 × 10^4^ km^2^. This reduction is particularly concentrated at the edges of medium-suitability areas, affecting regions such as Tibet, Sichuan, Inner Mongolia, and Shanxi. However, a shift occurs in the following decade, from 2041 to 2060 to 2061 to 2080, where the area of suitable habitat is anticipated to increase by 29.52 × 10^4^ km^2^. This expansion is primarily noted in Eastern Qinghai, most of Hebei, and Northern Henan. In contrast, the SSP5–8.5 scenario presents a different trend. From 2021 to 2040 to 2041 to 2060, the area of suitable habitat sees a modest increase of 2.48 × 10^4^ km^2^. This growth is mainly observed along the edges of suitable habitats in regions such as Beijing, Hebei, Ningxia, Gansu, Sichuan, and Yunnan. However, the period from 2041 to 2060 to 2061 to 2080 reveals a sharp decline in suitable habitat, with a decrease of 17.63 × 10^4^ km^2^. This change reflects a complex pattern, with both decreases and increases distributed across the edges of suitable habitats (Figure 2).

The high-aptitude areas for *mongolica* are primarily concentrated in the northwest. The medium and low aptitude areas are cross-distributed in this region, with natural boundaries such as the Kunlun Mountains, Qilian Mountains, Helan Mountain, and Mengyin Mountain. These areas are mainly located to the north of these mountain ranges, spanning across the provinces of Mongolia, Xinjiang, and Gansu. Under the SSP1–2.6 scenario, the suitable habitat area for *mongolica* Rousi decreases consistently across all periods, with the most significant reduction of 61.54 × 10^4^ km^2^ observed between 2041 and 2060. The primary areas of reduction include the Himalayan Mountain Range in Tibet, the Tarim and Junggar Basins in Xinjiang, and the vicinity of Helan Mountain in Inner Mongolia. In the SSP5–8.5 scenario, from 2061 to 2080, the largest reduction of suitable habitat is recorded, with a decrease of 38.82 × 10^4^ km^2^. The main areas of reduction are concentrated around the Altay Mountains in Xinjiang, the Central Tarim Basin, and the mountain ranges in Inner Mongolia (Figure 3).

The high suitability zone for *yunnanensis* is primarily located in the southwest of China, with the medium and low suitability zones scattered around Yunnan Province. Under the SSP1–2.6 scenario, from 2041 to 2060, the suitable habitat area decreases slightly by 8.57 × 10^4^ km^2^. This reduction continues from 2061 to 2080, with a further decrease of 13.16 × 10^4^ km^2^. These reductions are primarily concentrated in Gansu, Shaanxi, and Sichuan provinces. Under the SSP5–8.5 scenario, from 2041 to 2060, the trend reverses, showing an increase in suitable habitat by 16.98 × 10^4^ km^2^. This increase is predominantly observed in the northeastern part of Sichuan and the southeastern part of Tibet. However, from 2061 to 2080, the suitable area decreases in regions such as Gansu, Ningxia, Inner Mongolia, Xinjiang, Tibet, and Qinghai, with reductions occurring across various parts of these provinces (Figure 4).

The high suitability zone for *turkestanica* is primarily located in the northwestern part of China. The medium and low suitability zones are cross-distributed across the northwestern and western inland regions of China. Under the SSP1–2.6 scenario, from 2061 to 2080, the area of suitable habitat for *turkestanica* decreases sharply by 26.58 × 10^4^ km^2^. The main reductions are concentrated in the provinces of Gansu, Xinjiang, Ningxia, Sichuan, and Qinghai. Under the SSP5–8.5 scenario, from 2041 to 2060, the trend reverses, showing a significant increase of 33.97 × 10^4^ km^2^ in suitable habitat. This expansion is primarily observed along the edges of basins in Xinjiang, near the Tianshan Mountains, and in Tibet close to the Himalayas (Figure 5).

### 2.4. Migration of the Centroid of Suitable Habitats in Future Periods

ArcGIS was utilized to analyze the migration of the centroid of suitable habitats over time under the two emission scenarios, SSP1–2.6 and SSP5–8.5. The temporal and spatial distribution trends of the centroid of *sinensis* predominantly show a northeast or northwest migration (Figure 6). Under current climate conditions, the centroid of the suitable growth area for *sinensis* is primarily located in the southwest region of Gansu Province. Specifically, under the SSP1–2.6 emission scenario, this centroid migrates northeast to Southern Gansu in the 2030s and continues to shift northeastward in the 2050s, culminating in a net displacement of approximately 120 km. In contrast, under the SSP5–8.5 emission scenario, the centroid initially moves northeast to the Gannan area in the 2030s before shifting back southwest in the 2050s. Overall, while the centroid positions under both scenarios change relative to their current locations, they predominantly remain within the same general area (Table 4).

*H. rhamnoides* subsp. *mongolica* primarily migrates to the northwest or southwest (Figure 6). Currently, the centroid for *mongolica* is located in the south-central part of Xinjiang, approximately at coordinates 40° N and 85° E. Under the SSP1–2.6 scenario, this centroid gradually shifts northeastward in the 2030s, exhibiting a relatively significant movement. By the 2050s, the centroid redirects southwest, thereby reducing the distance from its initial position. Overall, the distribution remains relatively stable. Under the SSP5–8.5 scenario, the centroid starts in the northwest central part of Xinjiang, around 43° N and 88° E, and progressively migrates southwestward throughout the 2030s and 2050s (Table 4).

For *yunnanensis*, the centroid is primarily located near the Hengduan Mountains in the high-altitude region of Southwestern China. This subspecies mainly migrates to the southeast (Figure 6). Under the SSP1–2.6 scenario, the centroid gradually moves southeast over a long distance in the 2030s, followed by a slight southwest shift in the 2050s. Under the SSP5–8.5 scenario, the movement pattern remains similar to that observed under the low-emission scenario. The centroid shifts to the southwest and southeast in the 2030s and 2050s, respectively, though the overall movement distance is relatively short, indicating a stable distribution (Table 4).

The centroid of *turkestanica* is mainly concentrated in the Tarim River Basin. Similar to *yunnanensis*, this subspecies migrates mainly to the northwest or southwest (Figure 6). Under the SSP1–2.6 scenario, the centroid gradually shifts northeast in the 2030s, before moving southwest in the 2050s, overlapping with its current distribution with minimal change and displaying relative stability. Under the SSP5–8.5 scenario, the centroid of *turkestanica* migrates gradually northwestward, remaining primarily within the Tarim River Basin (Table 4).

## 3. Discussion

### 3.1. Effects of Environmental Factors on the Spatial Distribution of the Four Subspecies of Hippophae rhamnoides

#### 3.1.1. Effects of Environmental Factors on the Spatial Distribution of *sinensis*

The growth of plants is closely linked to hydrothermal conditions, with climate, precipitation, and temperature serving as critical factors influencing plant reproduction and growth [10]. It is predicted that *sinensis* is primarily distributed across the Qinghai–Tibet Plateau and the Western Sichuan Plateau, with additional, albeit smaller, populations located in Ningxia, Inner Mongolia, Hebei, and Liaoning, corroborating findings from previous studies [13]. The distribution of this species is influenced by a variety of environmental factors. The suitable range for the average temperature during the coldest quarter is between −6.24 and 3.59 °C, indicating that winter conditions in this region are relatively cold, though not extreme. Such climatic conditions are generally conducive to temperate or cold temperate ecosystems, characterized by vegetation predominantly composed of cold-resistant plants. *sinensis* demonstrates a robust adaptability to low temperatures, enabling it to maintain normal growth in a cold, yet non-extreme winter environment [14]. Annual precipitation in the region ranges from 506.38 to 1052.38 mm, suggesting favorable moisture conditions. Moreover, the actual climate data for the current distribution area aligns closely with these findings [13]. However, the suitable range of precipitation during the driest month is notably low, falling between 1.08 and 5.24 mm, indicating the presence of a distinct dry season. Nonetheless, the relatively high annual average precipitation suggests that either the dry season is brief or that precipitation is distributed unevenly throughout the year [26]. In this environment, the root system of *sinensis* is well developed, allowing it to efficiently absorb deep soil moisture. Additionally, the plant’s waxy leaves mitigate transpiration, a structural adaptation aligned with the predicted environmental factors [27].

Overall, the findings indicate that the climate of the regions suitable for *sinensis* is classified as either a temperate monsoon or continental climate, characterized by cold winters, moderate precipitation, and seasonal drought. The results derived from key environmental influencing factors are consistent with the actual distribution of suitable habitats.

#### 3.1.2. Effects of Environmental Factors on the Spatial Distribution of *mongolica*

In contrast to *sinensis*, the suitable habitats for *mongolica* are primarily located in the northwest regions of China, including Xinjiang, Inner Mongolia, and Gansu, which aligns with the literature records [20,28]. Among the key environmental factors influencing its distribution, the precipitation during the wettest month ranges from 31.78 to 33.61 mm, indicating that total annual precipitation in this region is extremely low. During the driest quarter (bio17), precipitation falls between 15.75 and 20.17 mm, which, while exceeding the extreme drought threshold of <5 mm, is concentrated in a brief rainy season; for instance, summer precipitation in the Mongolian Plateau accounts for more than 60% of annual rainfall [29,30]. The combination of drought conditions and intermittent precipitation likely drives *mongolica* to develop an efficient water utilization strategy, enabling it to rapidly absorb and store water during the rainy season. Research indicates that *mongolica* achieves effective replenishment through its deep root structure, with the main root extending 5 to 8 m deep and an increase in fine root density [31], as well as through a strategy of utilizing soil moisture stratification [32]. Furthermore, the soil moisture storage capacity of *mongolica* forests is relatively high, especially during the rainy season, allowing for effective rainwater storage for plant uptake and utilization [33]. The suitable habitat for *mongolica* is found at elevations ranging from 741.91 to 2222.73 m above sea level, which classifies the area as being of middle to high altitude, typically corresponding to a temperate continental plateau climate. This region is characterized by significant temperature fluctuations between day and night, a low average annual temperature, yet not extremely cold winters. *mongolica* adapts to the combined pressures of temperature and altitude through an optimization strategy that involves light energy and nutrient allocation [34].

In summary, the suitable habitat for *mongolica* may be classified as either a temperate semi-arid continental climate or a high-altitude arid mountain climate. This area exhibits marked vertical differentiation in precipitation, with lower altitudes experiencing drought conditions, while higher altitudes see slightly increased precipitation but are subject to strong evaporation. The predicted results align well with the actual distribution of suitable habitats.

#### 3.1.3. Effects of Environmental Factors on the Spatial Distribution of *yunnanensis*

*H. rhamnoides* subsp. *yunnanensis* is primarily distributed within a narrow zone at the intersection of Tibet, Sichuan, Yunnan, and Qinghai in China, exhibiting a limited distribution range. This observation is consistent with previous research findings [35]. The distribution of this subspecies is significantly constrained by the high-altitude mountain environment, with core environmental factors shaping its unique ecological niche. The altitude range for *yunnanensis* is between 2673.62 and 4018.12 m, corresponding to the cold temperate zone of alpine mountains and the subalpine climate zone [36]. The annual average temperature in this area is typically lower than 5 °C, with a notable temperature variation exceeding 20 °C between day and night. High-altitude exposure to strong ultraviolet radiation may compel *yunnanensis* to synthesize elevated concentrations of antioxidants, thereby mitigating oxidative stress [37]. The temperature seasonal coefficient of variation ranges from 532.92 to 642.78 (CV), which is substantially higher than that found in low-altitude temperate areas (where CV typically falls below 400). This indicates significant temperature fluctuations within the distribution area of *yunnanensis*, potentially linked to the interaction between the monsoon and plateau climates at the southeastern edge of the Qinghai–Tibet Plateau [38]. *H. rhamnoides* subsp. *yunnanensis* may adapt to these environmental conditions by regulating its growth cycle, such as through early leaf drop or a shortened dormancy period [39]. The average temperature range during the coldest quarter is between −2.99 and 5.09 °C, suggesting a periodic freeze–thaw cycle in winter, although sustained extreme low temperatures are absent. The plant likely employs physiological dormancy strategies, such as shedding leaves during winter and reducing metabolic activity to minimize water loss, alongside morphological adaptations such as shrub dwarfism (with plant heights typically less than 2 m) [40,41].

In conclusion, the suitable climate for *yunnanensis* can be classified as either an alpine cold temperate climate or a subalpine humid climate, aligning with the predicted climatic conditions for its suitable distribution area.

#### 3.1.4. Effects of Environmental Factors on the Spatial Distribution of *turkestanica*

The suitable distribution area of *turkestanica* is predominantly located in the inland arid-semi-arid transition zone of Northwestern China. Its ecological niche is significantly influenced by the environmental factor of precipitation during the wettest month, with a suitable range of 40.05 to 40.84 mm, characteristic of typical arid zones. The predicted suitable distribution area generally receives annual precipitation of less than 250 mm, particularly in regions such as the edges of the Tarim Basin in Xinjiang and the Hexi Corridor. Precipitation in these areas is concentrated within a brief summer period, aligning with the specified threshold range. Research indicates that vegetation experiencing wettest monthly precipitation of less than 50 mm typically adapts to moisture limitations through the development of deep root systems (exceeding 3 m) and efficient moisture utilization strategies, such as C4 photosynthesis [42,43]. *H. rhamnoides* subsp. *turkestanica* demonstrates resilience to low temperatures ranging from −16.79 to −8.39 °C. The predicted suitable distribution area, characterized as northwest inland, experiences severe cold in winter. For instance, the average temperature in January in Altay, Xinjiang, is approximately −15 °C, which closely corresponds to the estimated range of −16.79 to −8.39 °C, indicating key physiological mechanisms that facilitate its ability to survive winter conditions. Studies have shown that cold-resistant plants can lower their cell freezing points by accumulating osmotic regulatory substances, such as soluble sugars and proline [24]. Additionally, the average daily temperature difference in this region ranges from 12.41 to 14.04 °C, which is notable due to the minimal cloud cover and dry air typical of the northwest. This significant daily temperature variation enhances plant productivity by improving photosynthesis and reducing respiratory consumption at night; moreover, the photosynthetic rate of sea buckthorn has been shown to positively correlate with the daily temperature difference [44].

In summary, the estimated environmental thresholds for key bioclimatic factors (bio13, bio6, bio2) closely align with the actual distribution locations of *turkestanica*, particularly in the core areas of the northwest arid-semi-arid transition zone.

### 3.2. Potential Future Geographic Distribution of the Four Subspecies of Hippophae rhamnoides

#### 3.2.1. Analysis of Potential Geographical Distribution Trends of *sinensis*

The ecologically suitable core area for *sinensis* is primarily found in the eastern part of the Qinghai–Tibet Plateau and along the border of the Hengduan Mountains. This area benefits from stable hydrothermal conditions, characterized by an annual mean temperature fluctuation of less than 1.5 °C and annual precipitation ranging from 400 to 800 mm, providing a conducive ecological environment [45]. The core area features predominantly alpine meadow soils, with a pH between 6.5 and 7.5 and organic matter content greater than 3%, exhibiting a high level of synergy with nitrogen-fixing bacteria symbiotic systems [46,47]. Long-term environmental pressures have led to the selection of ecotypes that benefit from terrain shielding effects, thereby maintaining the diversity of populations within the core area through mechanisms of epigenetic regulation.

However, under the current climate conditions, marginal areas of species distribution, such as the southern sections of the Hengduan Mountains and the Mengyin Mountains, are experiencing a retreat. This habitat retreat is attributed to the combined effects of climate change and human activities. Research has indicated that under the RCP4.5 scenario, the annual mean temperature in these marginal zones is projected to rise by 1.5 to 2.0 °C, intensifying summer droughts and altering precipitation patterns. These changes are likely to lead to increased soil erosion, which negatively impacts seedling settlement and growth [48,49]. Additionally, agricultural expansion and grazing pressure in low-altitude river valleys have resulted in local extinction rates of up to 45% [50]. Notably, under the SSP585 scenario, a phased expansion–contraction dynamic is observed. Annual precipitation in North China is expected to increase by 10 to 15%, and winter temperatures are projected to rise, temporarily extending the northern survival boundary of sea buckthorn. As original vegetation, such as temperate grasslands, degrades due to drought, sea buckthorn is predicted to occupy these ecological niches through a pioneer species strategy [50,51]. From 2040 to 2080, the area of the suitable zone decreased sharply, and the distribution of the suitable zone edge was reduced. In the late SSP585 period, the number of high-temperature days in summer increased by 20 days, the precipitation variability increased, and the water stress index exceeded the tolerance limit. The marginal population had low genetic diversity, and the adaptive evolution rate was insufficient to cope with rapid climate change, resulting in a sharp decrease in the area of the suitable zone [52,53].

#### 3.2.2. Analysis of Potential Geographical Distribution Trends of *mongolica*

The core distribution area of *mongolica* is concentrated in the arid-semiarid transition zone in Northwest China, including the northern foothills of the Kunlun Mountains, the eastern section of the Qilian Mountains, the Helan Mountains, and the area north of the Yinshan Mountains. The stability of the core area is maintained by topographic–climate synergy, soil adaptability, and genetic conservatism. The Kunlun Mountains and the Qilian Mountains block the dry, hot air flow, forming a local humid microclimate (annual average temperature 8–9 °C), which alleviates drought stress [54]. The diversity of the shady slopes at an altitude of 1700–1800 m in the northern section of the Helan Mountains is the lowest, while the shady slopes at an altitude of 2000–2100 m in the middle section have become the dominant area for sea buckthorn due to stable climatic conditions [55]. In addition, the soil in the core distribution area is mainly gray–calcic soil and brown–calcic soil, with a pH of 7.5–8.5 and an organic matter content of 0.77–1.28 g/kg. Nitrogen utilization is enhanced by symbiotic nitrogen-fixing bacteria (Frankia spp.) [56]. The niche conservatism index of *mongolica* to the historical climate (LGM period) limits its spread to new habitats [57].

Under the SSP126 scenario, the area of the suitable habitat of *mongolica* in all periods has shrunk, and the reduced area is distributed in the Himalayas in Tibet, the Tarim Basin in Xinjiang, the Junggar Basin, and the vicinity of the Helan Mountains in Inner Mongolia. The trend of change in the marginal area is speculated to be caused by precipitation decrease and warming. The annual precipitation in the Tarim Basin has decreased by 10–15%, exceeding the survival threshold of *Hippophae rhamnoides* L. [58]. Winter warming in the Himalayan marginal area has led to unstable snow cover, and the freeze–thaw cycle has damaged the root system of seedlings [59]. Under the SSP585 scenario (high emissions), the suitable habitat dynamics for *mongolica* demonstrate both contraction and expansion areas. The contraction area is in the southern foothills of the Altai Mountains, the center of the Tarim Basin, and the Inner Mongolia Mountains, and the expansion area is in Northern Xinjiang (west of the Altai Mountains). Studies have shown that extreme drought conditions in the Tarim region will intensify, and water stress will lead to population decline [60]. At the same time, under the high emission scenario, the Northern Xinjiang region will experience warming and wetting, with warmer winters, breaking through the northern limit [61].

#### 3.2.3. Analysis of Potential Geographical Distribution Trends of *yunnanensis*

*H. rhamnoides* subsp. *yunnanensis* is a subspecies endemic to China. Its core distribution area is located in the Hengduan Mountains in Southwestern China and the northern edge of the Yunnan–Guizhou Plateau. This distribution pattern is closely related to the high mountain and canyon terrain of the Hengduan Mountains running north–south [62]. The Hengduan Mountains are one of the origin centers of the genus *Hippophae*. The complex terrain and monsoon climate provide a stable ecological niche for Yunnan Sea buckthorn [63]. Under the low emission pathway (SSP126), the northern suitable habitat area of *yunnanensis* (such as north of the Qinling Mountains and Huai River) will face significant shrinkage, with a reduction of about 15% in the suitable area from 2041 to 2060. In the SSP585 scenario, the trend is opposite, with the suitable area for *yunnanensis* increasing by about 12% from 2041 to 2060, with the expansion core located in the Qilian Mountains (2800–3800 m) [24]. The dynamics of the marginal area next to the core area are driven by changes in precipitation, temperature, and soil conditions in future climate scenarios. Contraction under the SSP126 scenario (low emissions). The warming of the northern winter breaks the low-temperature dormancy requirement of *yunnanensis*, resulting in spring phenological disorder and a 40% increase in the flowering failure rate [64]. The shrinking areas were in low-altitude areas such as Southern Gansu, Liupan Mountain in Ningxia, and Yinshan Mountain in Inner Mongolia, where high temperatures in summer caused a 50% decrease in photosynthetic enzyme activity [65].

The vertical migration ability of *yunnanensis* is constrained by the synergistic effect of terrain and soil. The current upper limit of altitude in the core distribution area (3500 m) may move up to 4000 m, and some areas of the Hengduan Mountains (such as the eastern slope of Gongga Mountain) may form new bare land due to glacier retreat. The pH value of the soil in the early stage of development is relatively low (6.5–7.0), which is conducive to the colonization of nitrogen-fixing bacteria [66,67]. The downward migration obstacles mainly come from the soil adaptability in the low-altitude area. For example, the clay area in the northern edge of the Yunnan–Guizhou Plateau leads to a 40% decline in the seedling emergence rate and a 60% reduction in the root–shoot ratio in the aeolian sandy soil environment [68]. In addition, warming promotes the expansion of subtropical evergreen broad-leaved forests towards higher altitudes, compressing the ecological niche width of *yunnanensis* [66].

#### 3.2.4. Analysis of Potential Geographical Distribution Trends of *turkestanica*

The core distribution area of *turkestanica* is also concentrated in the arid–semiarid transition zone in Northwest China, covering the northern edge of the Tarim Basin in Xinjiang (southern foot of the Tianshan Mountains), the southern edge of the Junggar Basin (northern slope of the Kunlun Mountains), the Central and Western Hexi Corridor in Gansu (eastern section of the Qilian Mountains), and the southwestern part of the Alxa Plateau in Inner Mongolia. This stable distribution pattern is closely related to the geomorphic transition zone and groundwater distribution in the northwest region [69]. When soil moisture decreases, *turkestanica* turns to deep soil water (utilization rate reaches 78.8%), while *turkestanica* directly uses ancient riverbed groundwater and tolerates an environment with an average annual precipitation of <200 mm [32,70]. During the Last Glacial Maximum (LGM), the Tarim Basin maintained a continuous suitable habitat due to topographic barriers, avoiding the population bottleneck effect. Similar studies on Adiantum nelumboides have shown that LGM refuges play a key role in the preservation of genetic diversity [71].

Under the SSP126 scenario, the area of the suitable habitat of *turkestanica* has been declining significantly, with the main areas of reduced area distributed in the eastern section of the Hexi Corridor in Gansu and central Ningxia. The CMIP6 model of the Yellow River Basin predicts that under the SSP126 scenario, enhanced evaporation will cause soil EC > 10 ds/m, exceeding the upper limit of HrNHX1 gene expression [72]. In addition, the intensification of salinization and the reduction of precipitation form positive feedback. The groundwater level in the Alexa Plateau dropped by >5 m, exceeding the root reachable depth, leading to population decline [73]. Under the SSP585 scenario, the trend is the opposite. The area of suitable area will increase sharply from 2041 to 2060. The increase in area is mainly distributed in the basin edge and near the Tianshan Mountains in Xinjiang. In addition, there is also an increase near the Himalayas in Tibet. The dominant environmental factor affecting *turkestanica* is bio13 (precipitation in the wettest month), while bio13 on the western edge of the Tarim Basin increases to 60–70 mm, and short precipitation pulses meet the germination requirements [74].

### 3.3. Migration Trend of the Center Point of Suitable Habitat for Sea Buckthorn Under Climate Change

Under the SSP1–2.6 emission scenario, the centroid of *sinensis* has migrated from areas with low annual precipitation to regions characterized by higher annual precipitation. This trend aligns with our primary environmental factor analyses, indicating that precipitation is a crucial determinant influencing the distribution of suitable habitats for *sinensis*. Consequently, suitable habitats are gradually shifting towards areas with improved precipitation conditions. Supporting this trend is the warming and humidification transition observed in Northwest China since the 1980s, which includes an average annual temperature increase of 0.7 °C and a precipitation rise of 10–20% [75].

In contrast, under the SSP5–8.5 emission scenario, the distribution area of the centroid of *sinensis* shows minimal change; however, the impact of climate change on suitable habitats becomes more pronounced. This results in erratic centroid migration patterns, likely due to unstable precipitation regimes and significant temperature increases. For example, while annual precipitation in the eastern part of Northwest China has decreased, the western part has experienced increases, leading to regional disparities in ecological responses [76]. Moreover, the increasing frequency of extreme precipitation events, such as short-term heavy rainfall, raises the risk of soil erosion and may compromise the root structure of *sinensis* [77]. This complexity underscores that under high emission scenarios, the influence of climate change on suitable habitats is both multifaceted and significant. Overall, the centroid distribution points of *sinensis* do not exhibit large-scale migration, suggesting the species has an inherent ability to adapt swiftly to environmental changes.

For *mongolica*, under the SSP1–2.6 scenario, the centroid migrates to areas where annual precipitation ranges between 150 and 300 mm, positioning it within relatively arid regions. The overall stability of its centroid distribution points can be attributed to the species’ specialized adaptation strategies in these arid environments. Research indicates that *mongolica* effectively responds to fluctuations in precipitation by adjusting its branch biomass allocation and optimizing soil water use [32,78]. Under the SSP5–8.5 scenario, the centroid migrates southwest, potentially as a response to high-temperature stress affecting its ecological niche. This directional shift may reflect the species’ adaptation to intensifying drought conditions, such as the expansion of desertification on the fringes of the Tarim Basin, which could lead to habitat contraction [79].

The stability of *yunnanensis*, particularly in high-altitude areas (with similar centroid changes under both SSP1–2.6 and SSP5–8.5 scenarios), can be linked to its unique mechanisms for stress resistance. High-altitude populations enhance their resistance to strong radiation by increasing the absorption of ultraviolet substances and elevating antioxidant enzyme activities (SOD, CAT). They also capitalize on concentrated rainfall during the rainy season to sustain growth. Additionally, the root system’s soil fixation capability on 30°~35° slopes (with a root-to-crown ratio reaching 0.8) further contributes to habitat stability [80]. This adaptive strategy contrasts with that of *turkestanica*, which relies more heavily on groundwater recharge in the Tarim River Basin [81,82].

*H. rhamnoides* subsp. *turkestanica* predominantly inhabits the Tarim River Basin, where average annual precipitation ranges from 25 to 50 mm, placing it in a highly arid environment. Under the SSP1–2.6 scenario, suitable habitats for *turkestanica* extend northeastward, potentially linked to ecological water transfer projects in the Tarim River Basin. Studies have shown that reducing groundwater depth from 8 m to 4 m can increase natural vegetation coverage by 30%, facilitating the spread of sea buckthorn to riparian zones [81,83]. In the context of the SSP5–8.5 scenario, the centroid of *turkestanica* gradually migrates northwest, indicating that as extreme climate conditions intensify, suitable habitats in arid areas may face greater pressures and exhibit more pronounced migration trends. The observed “contraction-redistribution” pattern resulting from extreme drought suggests that although the total area of suitable habitats may decrease, local oases (such as the Aksu River Basin) are likely to serve as refuges due to increased glacial meltwater [84].

## 4. Methods

### 4.1. Collection and Analysis of Data on the Distribution of Hippophae rhamnoides

#### 4.1.1. Acquisition of Data Points on the Distribution of *Hippophae rhamnoides* in China

By consulting the Flora of China (1983 edition), the Digital Herbarium of China (Institute of Botany, Chinese Academy of Sciences, 2004–2020; http://www.cvh.org.cn; accessed 15 January 2025), the Global Biodiversity Information Platform (GBIF.org, 2020; www.gbif.org; accessed 10 May 2025), the National Herbarium Resource Platform of China (NSII, 2020; www.nsii.org.cn; accessed 10 May 2025), and the National Specimen Information Infrastructure (Figure 7), we finally organized and obtained 1887 distribution points (Figure 8).

#### 4.1.2. Environmental Data Collection and Analysis

Bioclimatic factors for the future periods (2021–2040, 2041–2060, 2061–2080, 2081–2100) were obtained from the World Climate Database (WorldClim, 2000–2024; www.worldclim.org; accessed 15 January 2025) for two climate scenarios (SSP1–2.6 and SSP5–8.5), covering 19 bioclimatic factors. These variables, derived from annual trends, seasonality, and minimum and maximum temperature and precipitation values, have significant biological relevance [85]. Climate data were sourced from global climate models (GCMs), specifically the BCC-CSM2-MR model from the Beijing Climate Center, with a resolution of 2.5 min (approximately 5 km geographic resolution). BCC-CSM2-MR has been shown to effectively reproduce precipitation and temperature patterns in East Asia, offering higher accuracy and reliability than its predecessor, BCC-CSM1.1m [86].

For future climate scenarios, four greenhouse gas emission pathways (SSP1–2.6, SSP2–4.5, SSP3–7.0, and SSP5–8.5) were initially considered. For this study, we selected two representative scenarios recently released by the IPCC (Intergovernmental Panel on Climate Change): SSP1–2.6 and SSP5–8.5 [87]. SSP1–2.6 and SSP5–8.5 correspond to the 5th and 95th percentiles of temperature rise in IPCC AR6, respectively, capturing the full range of bioclimatic variable changes. They represent the lowest and highest levels of radiative forcing in global climate change scenarios [88,89]. Comparing the two reveals the differential impacts of climate policy intervention (SSP1–2.6) versus “business as usual” (SSP5–8.5) on species distribution, providing decision-makers with a “best-to-worst” reference framework [90,91]. Listed as baseline scenarios by CBD and IPBES, SSP1–2.6 aligns with NDCs for emission reductions, while SSP5–8.5 warns of non-compliance risks [89,92].

### 4.2. Species Distribution Data

The number of environmental variables significantly impacts the complexity of the MaxEnt model [93]. Reducing the number of climate predictor variables and selecting the most relevant ones are crucial for improving the transferability of the MaxEnt model across different climate scenarios [94]. In this study, both climate and topographic variables were considered as environmental factors potentially influencing the distribution of *Hippophae rhamnoides* species. These variables were imported into MaxEnt version 10.8.1 for preliminary simulations. Contribution rates of each variable to habitat suitability were calculated, and those with rates < 10% (threshold: 0.1) were excluded from subsequent analyses.

#### Processing with ENMTools

ENMTools version 1.0.1 is a software tool designed for the analysis of ecological niche models, enabling quantitative analysis of ecological niches, similarity measures, and statistical tests, while also interacting with MaxEnt [95]. ENMTools was utilized to analyze the impact of environmental factors by calculating the correlation (R-values) between them. Factors with R-values greater than 0.8 were compared in terms of their contribution to the species’ distribution. Ultimately, the factor with the highest contribution was selected for further use in the model [96].

### 4.3. MaxEnt Modeling

Based on the results of the preliminary experiment, the most influential environmental variables were selected for the MaxEnt model. The relevant ASC format files were named in a standardized manner. The output from MaxEnt software, consisting of raster layers in ASC format, provided a fitness index ranging from 0 to 1. These results were classified into four distinct fitness classes based on the index, using the Reclassify Tools in ArcGIS’s Spatial Analyst module. The results are shown in different colors. The prediction results were then analyzed and visualized using ArcGIS software, producing potential distribution maps of *Hippophae rhamnoides* under two different climate scenarios, SSP1–2.6 (low emissions) and SSP5–8.5 (high emissions). The actual geographic distribution of *Hippophae rhamnoides* in China was mapped in ArcGIS based on distribution data, elevation data, and vegetation type data. Using ArcGIS’s overlay functions (ArcMap-Spatial Analyst-Raster Calculator), the potential distribution areas of *Hippophae rhamnoides* were predicted and visualized. The final distribution patterns of species diversity for *Hippophae rhamnoides* under the two climate scenarios were also derived.

### 4.4. Reliability Assessment of MaxEnt Modeling

In this study, the data were analyzed using MaxEnt version 3.4.4 model prediction software and ArcGIS 10.8.1 software. The processed distribution data of *Hippophae rhamnoides* in CSV format and climate variables in ASC format were imported into MaxEnt. The dataset was randomly partitioned, with 75% of occurrence points used for model training and the remaining 25% for testing. The model was run with 10 replicates, and MaxEnt automatically generated the Receiver Operating Characteristic (ROC) curve and computed the Area Under the Curve (AUC) value [97]. The AUC values were derived using both parametric and non-parametric methods. In all cases, the AUC values exceeded 0.9, indicating model stability [25]. Theoretically, an AUC value between 0.5 and 0.7 suggests poor predictive performance, while values between 0.7 and 0.9 indicate moderate predictive capability. An AUC exceeding 0.9 signifies high predictive accuracy, rendering the model results reliable and robust [98].

### 4.5. Identification of Suitable Habitats and Center-of-Mass Analysis

The mean habitat suitability output from the 10 MaxEnt replicates (in ASC format) was imported into ArcGIS and converted into a shapefile (SHP format). The natural breaks (Jenks) classification method was applied using the reclassification tool to categorize habitat suitability into four classes: unsuitable, low, moderate, and high suitability zones. The area of each suitability category was then quantified. The Jenks method optimizes the inter-class variance to adapt to the distribution characteristics of the data itself, especially suitable for scenarios where the habitat suitability index has a non-uniform distribution [99]. In the literature, MaxEnt outputs are commonly classified into four categories (such as high/medium/low/unsuitable), which match the response patterns of species to environmental gradients [100,101]. The common criteria for MaxEnt model include high suitability (≥0.6), moderate suitability (0.3–0.6), low suitability (0.1–0.3), and unsuitability (<0.1). The deviation between the natural fracture result and the theoretical threshold is less than 10% and does not need to be modified [100,102].

## 5. Conclusions

Future climate change poses an escalating challenge due to increasing drought conditions, necessitating urgent and effective measures for mitigation. This study employed the MaxEnt model and ArcGIS software to simulate the potential geographical distribution of four subspecies of *Hippophae rhamnoides* across current and future periods (2021–2040, 2041–2060, and 2061–2080). The spatial distribution and migration trends of these subspecies illustrate the ecological filtering effects resultant from the interplay of precipitation, temperature, and geographical environment.

*H. rhamnoides* subsp. *mongolica* and *sinensis* predominantly inhabit arid and semi-arid regions, leveraging groundwater utilization, precipitation pulse synchronization, and root adaptations to sustain their drought resilience. In contrast, *yunnanensis*, the only high-altitude subspecies, adapts to climatic fluctuations through vertical migration and ultraviolet-induced metabolic processes. Its genetic diversity markedly differs from that of the other subspecies, highlighting the evolutionary significance of the Hengduan Mountains as a biological refuge.

The distribution patterns of the four subspecies reveal a “convergence and divergence” framework under climate stress, where climatic factors shape ecological adaptability through natural selection. These adaptive features, in turn, restrict geographic distribution boundaries, resulting in distinct core suitable areas characterized by niche conservatism. This conservatism stabilizes core areas, while phenotypic and genetic plasticity enables adaptation in marginal distributions to meet climate challenges.

The findings present a model for understanding species response mechanisms in arid and alpine environments. However, they also caution that future extreme climatic events could surpass adaptive thresholds through phenomena such as groundwater depletion and increased UV-B radiation. There is an urgent need to integrate climate, ecological, and genetic data into conservation strategies to promote precise and proactive biodiversity preservation.

## Figures and Tables

**Figure 1 plants-14-01682-f001:**
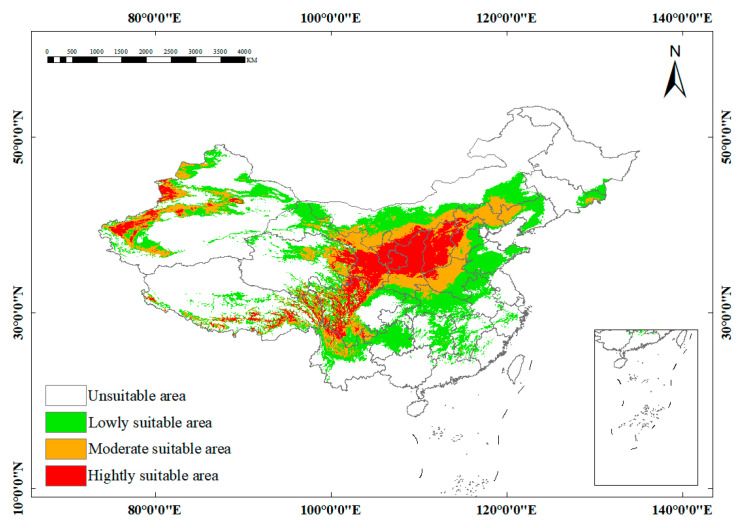
The suitable land distribution areas for four species of *Hippophae rhamnoides* under current climatic conditions.

**Figure 2 plants-14-01682-f002:**
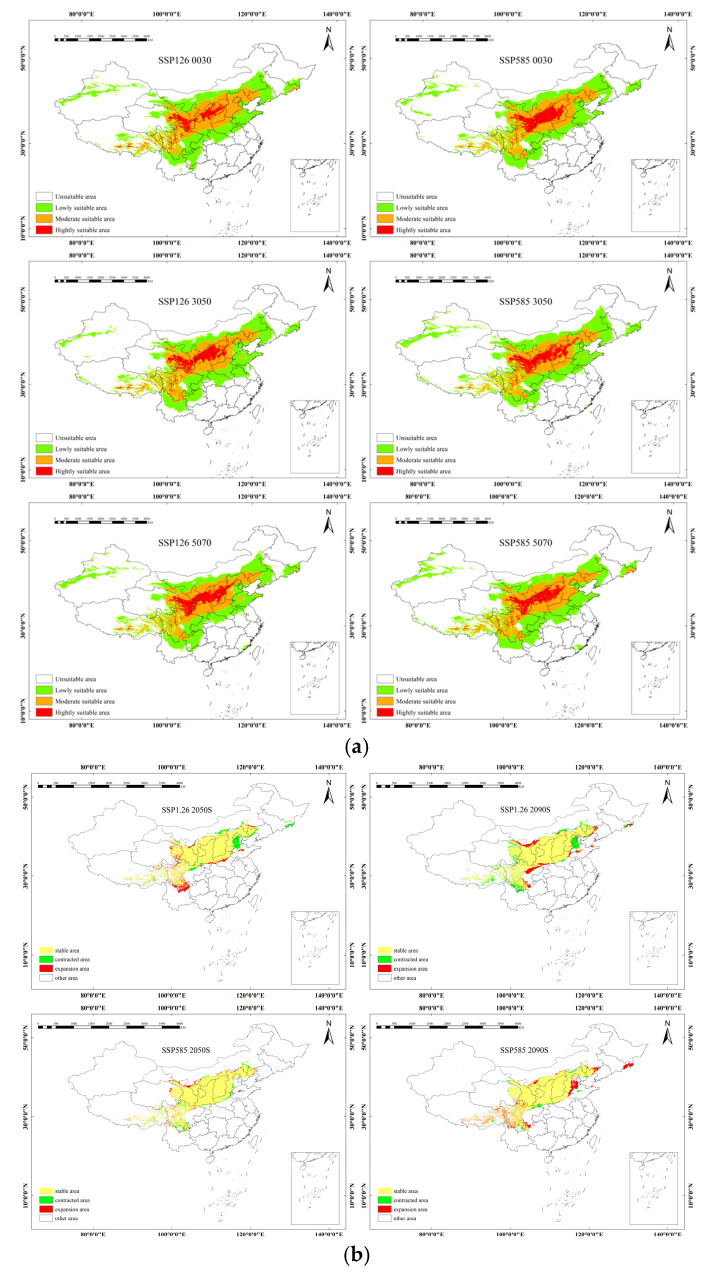
(**a**) The suitable distribution areas and contraction expansion of *sinensis* under two climate models for future climate scenarios. (**b**) Spatial pattern changes in suitable distribution for *sinensis* under future climatic conditions.

**Figure 3 plants-14-01682-f003:**
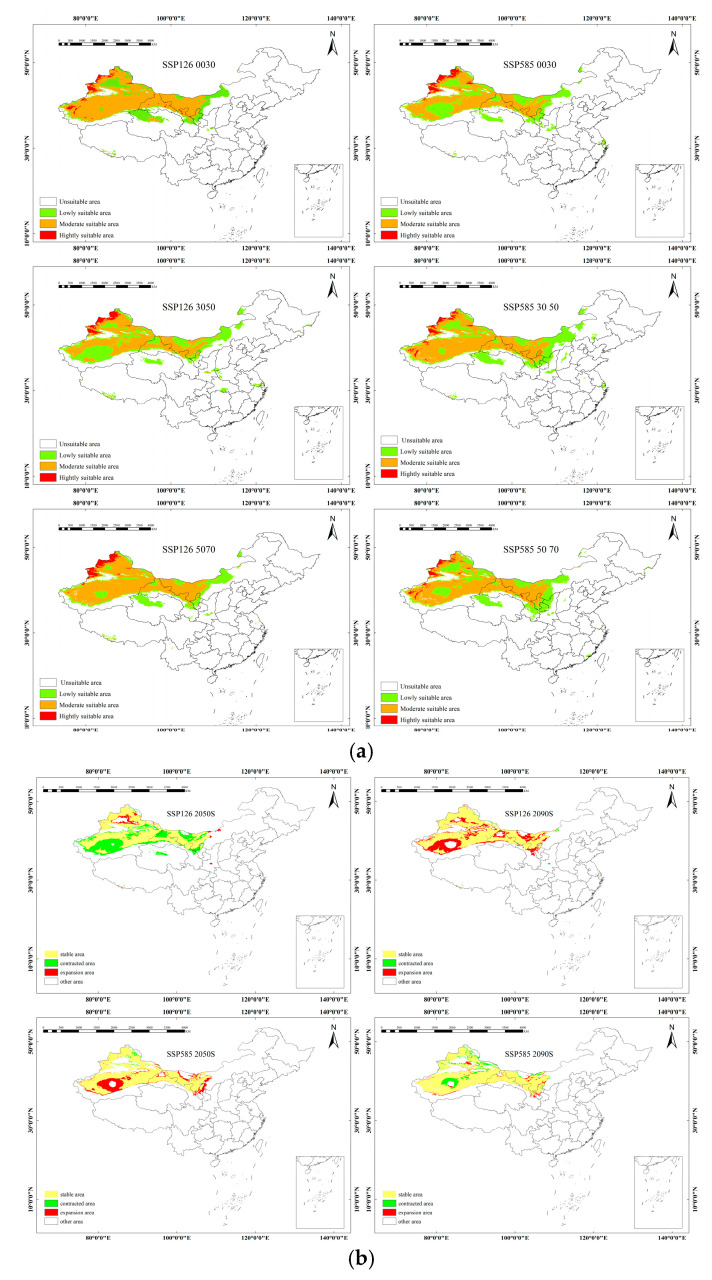
(**a**) The suitable distribution areas for *mongolica* under the future climate scenarios of the two climate models. (**b**) Spatial pattern changes in suitable distribution for *mongolica* under future climatic conditions.

**Figure 4 plants-14-01682-f004:**
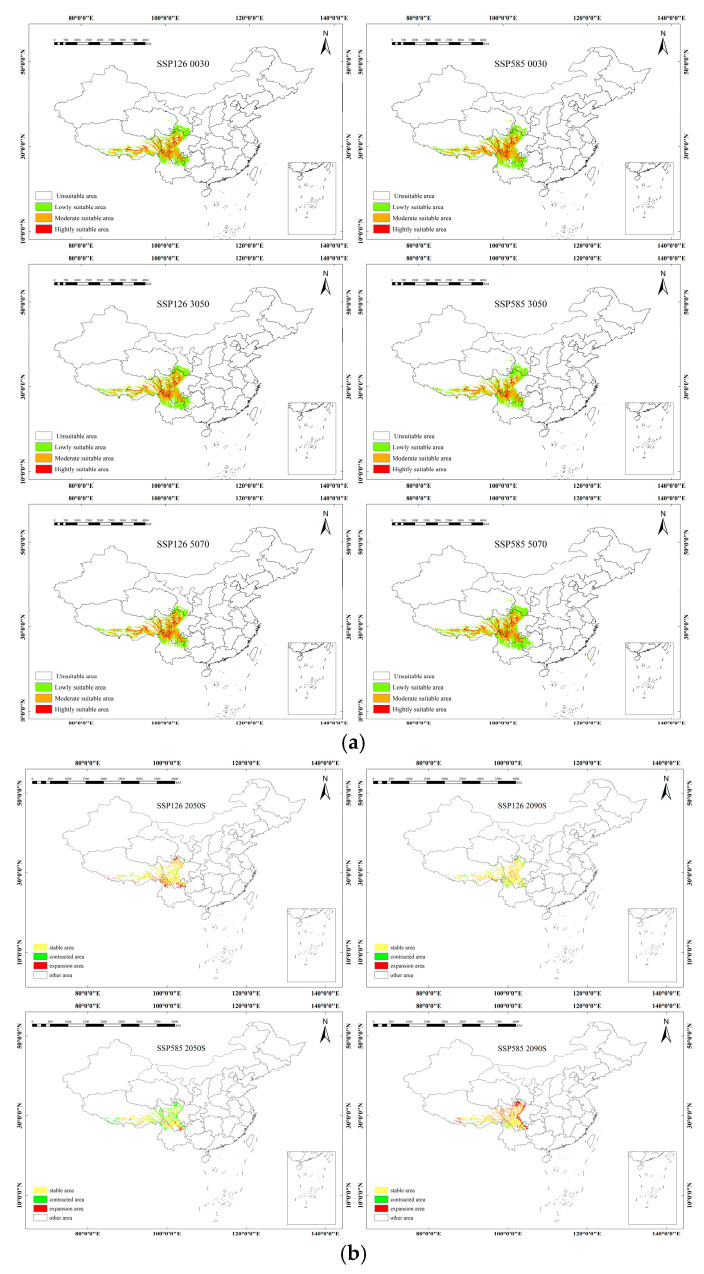
(**a**) The suitable distribution areas for *yunnanensis* under future climate scenarios with two climate models. (**b**) Spatial pattern change in suitable distribution for *yunnanensis* under future climatic conditions.

**Figure 5 plants-14-01682-f005:**
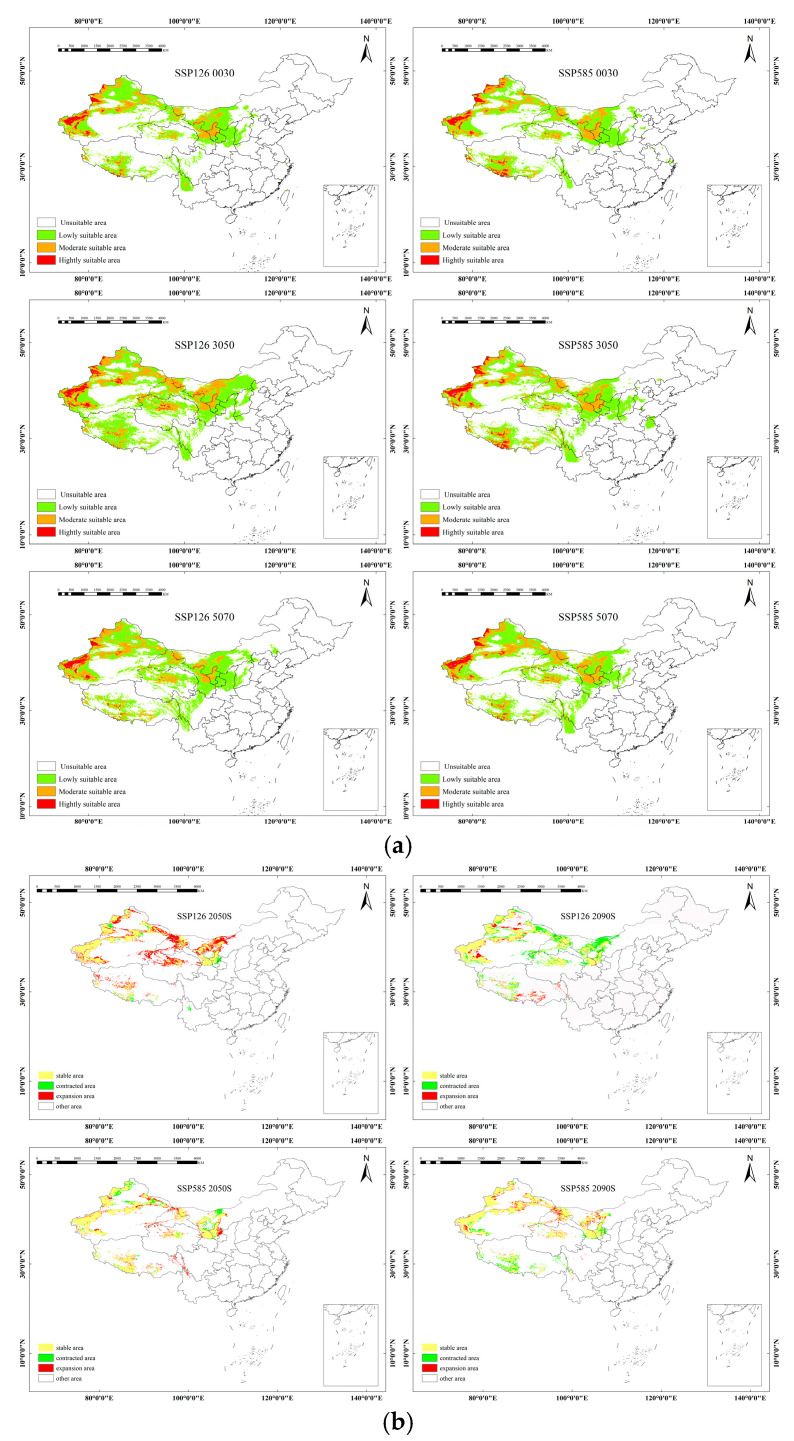
(**a**) The suitable distribution areas for *turkestanica* under future climate scenarios with two climate models. (**b**) Spatial pattern changes in suitable distribution for *turkestanica* under future climatic conditions.

**Figure 6 plants-14-01682-f006:**
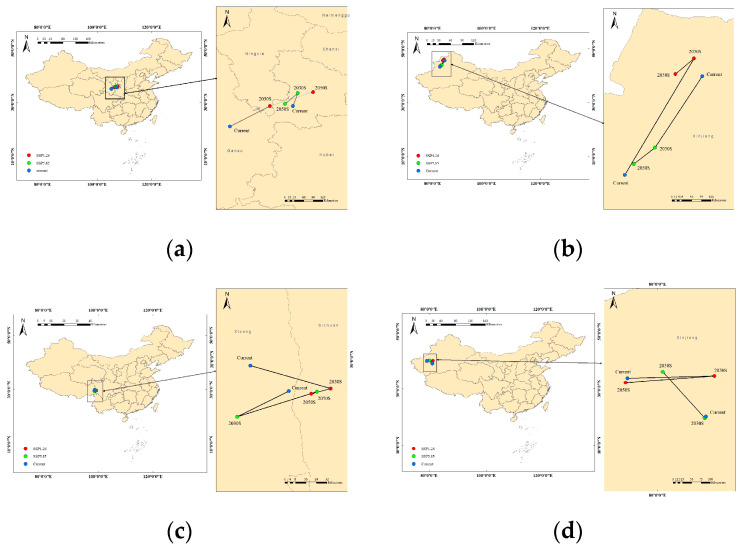
Migration path of *Hippophae rhamnoides* suitable habitat centers under current and future climate scenarios. (**a**) *sinensis* (**b**) *mongolica* (**c**) *yunnanensis* (**d**) *turkestanica*.

**Figure 7 plants-14-01682-f007:**
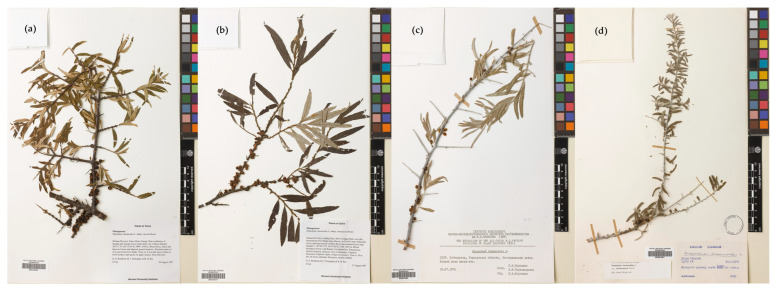
Quoted from the specimen photos of four subspecies of sea buckthorn on the Global Biodiversity Information Platform: (**a**) *sinensis*; (**b**) *yunnanensis*; (**c**) *mongolica*; (**d**) *turkestanica*.

**Figure 8 plants-14-01682-f008:**
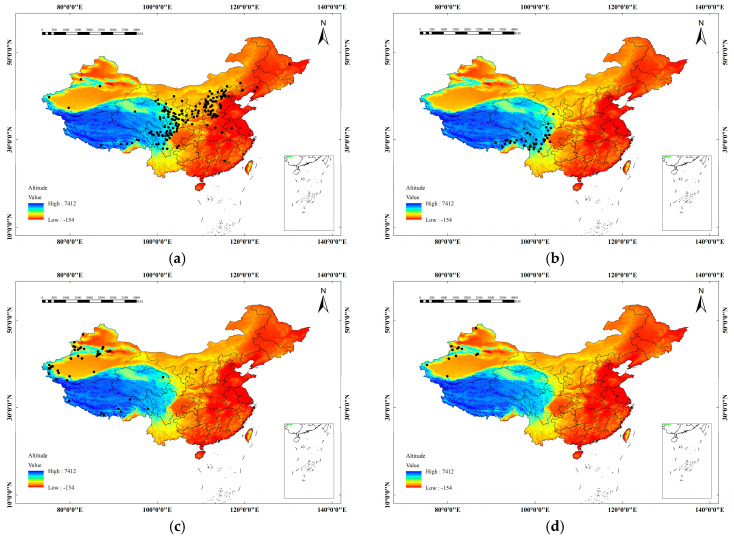
Spatial distribution of *H. rhamnoides* in China. (**a**) *sinensis*; (**b**) *yunnanensis*; (**c**) *mongolica*; (**d**) *turkestanica*.

**Table 1 plants-14-01682-t001:** Contribution rate and performance evaluation of MaxEnt model environment variables.

Ranking	Environment Variable	Full Name (In English)	Contribution Rate (%)
1	bio11	Mean Temperature of Coldest Quarter	20.0%
2	Altitude	Elevation	18.7%
3	bio6	Min Temperature of Coldest Month	11.6%
4	bio12	Annual Precipitation	9.2%
5	bio2	Mean Diurnal Range (Mean of Monthly Max–Min Temperature)	7.8%
6	bio15	Precipitation Seasonality (Coefficient of Variation)	4.4%
7	bio17	Precipitation of Driest Quarter	4.4%
8	bio4	Temperature Seasonality (Coefficient of Variation)	4.4%
9	bio7	Annual Temperature Range (Max Temperature of Warmest Month − Min of Coldest)	4.0%
10	bio14	Precipitation of Driest Month	4.0%

**Table 2 plants-14-01682-t002:** Comparison of dominant environmental factors and suitable range of four subspecies of *Hippophae rhamnoides*.

Subspecies	Dominant Environmental Factors	Full Name (In English)	Suitable Range
*sinensis*	bio12	Annual Precipitation	506.38–1052.38 mm
	bio14	Precipitation of Driest Month	1.08–5.24 mm
	bio11	Mean Temperature of Coldest Quarter	−6.24–3.59 °C
*mongolica*	bio13	Wettest monthly precipitation	31.78–33.61 mm
	Altitude	Elevation	741.91–2222.73 m
	bio17	Precipitation of Driest Quarter	15.75–20.17 mm
*yunnanensis*	Altitude	Elevation	2673.62–4018.12 m
	bio4	Temperature Seasonality	532.92–642.78
	bio11	Mean Temperature of Coldest Quarter	−2.99–5.09 °C
*turkestanica*	bio13	Wettest monthly precipitation	40.05–40.84 mm
	bio2	Mean Diurnal Range (Mean of Monthly Max–Min Temperature)	12.41–14.04 °C
	bio6	Min Temperature of Coldest Month	−16.79–−8.39 °C

**Table 3 plants-14-01682-t003:** The area of each habitat of four *Hippophae rhamnoides* subspecies in current distribution analysis of potentially suitable habitat.

Subspecies	Total Suitable Habitat Area	Percentage of China’s Land Area	Main Distribution Regions	Highly Suitable (Proportion)	Moderately Suitable (Proportion)	Low-Suitability (Proportion)
*sinensis*	3.0142 × 10^6^ km^2^	31%	Central China (Qinghai–Tibet–Sichuan border regions); scattered in Ningxia, Inner Mongolia, Hebei, Liaoning	0.6203 × 10^6^ km^2^ (21%)	0.6656 × 10^6^ km^2^ (22%)	1.753 × 10^6^ km^2^ (58%)
*mongolica*	3.0220 × 10^6^ km^2^	31%	Northwestern China (Xinjiang, Inner Mongolia, Gansu); scattered in Shanxi, Tibet, Hebei	0.3434 × 10^6^ km^2^ (11%)	1.5712 × 10^6^ km^2^ (52%)	1.1321 × 10^6^ km^2^ (37%)
*yunnanensis*	0.9555 × 10^6^ km^2^	10%	Narrow belt along Tibet–Sichuan–Yunnan–Qinghai borders	0.1704 × 10^6^ km^2^ (18%)	0.2508 × 10^6^ km^2^ (26%)	0.5589 × 10^6^ km^2^ (58%)
*turkestanica*	2.7403 × 10^6^ km^2^	29%	Xinjiang (Kashgar, Yining, Hotan); Tibet, Ningxia, Gansu, Qinghai, Inner Mongolia	0.2999 × 10^6^ km^2^ (11%)	0.8193 × 10^6^ km^2^ (30%)	1.6457 × 10^6^ km^2^ (60%)

**Table 4 plants-14-01682-t004:** Summary of changes in suitable habitats and centroid migration for four subspecies of sea buckthorn in the future.

Subspecies	Climate Scenario	Habitat Area Change (×10^4^ km^2^)	Key Time Period	Direction and Distance of Centroid Migration
*sinensis*	SSP1–2.6	−37.41 (2021–2060) → +29.52 (2061–2080)	2021–2060	Northeast migration, accumulated about 120 km
	SSP5–8.5	+2.48 (2021–2060) → −17.63 (2061–2080)	2061–2080	Swinging from northeast to southwest, generally stable in Southern Gansu
*mongolica*	SSP1–2.6	−61.54 (2041–2060)	2041–2060	Swinging from northeast to southwest, generally stable in central and Southern Xinjiang
	SSP5–8.5	−38.82 (2061–2080)	2061–2080	Continuously migrating southwest and ultimately staying in the northwest of Xinjiang
*yunnanensis*	SSP1–2.6	−8.57 (2041–2060) → −13.16 (2061–2080)	2041–2080	Short distance migration from southeast to southwest, overall stable
	SSP5–8.5	+16.98 (2041–2060) → Widespread reduction (2061–2080)	2041–2060	Short distance migration from southwest to southeast
*turkestanica*	SSP1–2.6	−26.58 (2061–2080)	2061–2080	Swing from northeast to southwest, stay in Tarim Basin
	SSP5–8.5	+33.97 (2041–2060) → subsequent reduction	2041–2060	Continuous northwest migration, staying in Tarim Basin

Symbol description: Negative value (−): Decrease in habitat area or contraction of center of mass towards migration direction. Positive value (+): Habitat area expands or center of mass expands towards migration direction.

## Data Availability

The original contributions presented in the study are included in the article, further inquiries can be directed to the corresponding authors.

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
