# Peer review of "Analysis of Spatial Suitable Habitats of Four Subspecies of Hippophae rhamnoides in China Based on the MaxEnt Model"

_plants, 2025, doi:10.3390/plants14111682_

Round 1
Reviewer 1 Report
Comments and Suggestions for Authors
Reviewer(s)' Comments to Author:
Manuscript ID: plants-3608009
Title: Analysis of spatial suitable habitats of four subspecies of Hippophae rhamnoides in China based on the MaxEnt model
Authors: MengYao He, FanYan Ma, JunJie Ding, PanXin Niu, CunKai luo, Mei Wang, Ping Jiang
First, congratulations to the authors, very nice article, complex work. Predicting and modelling the impact of weather parameters on the potential distribution area of different species is always an important and interesting research topic.
However, it would be worthwhile to better present the species and subspecies that are the subject of the study.
The differences between the individual subspecies appear in the discussion, but the entire research would be better understood if the most important morphological and physiological differences were presented in the methods chapter. I would appreciate a schematic draw or photograph about the studied species and subspecies.
In addition, the extremely detailed results should also be better summarized. That is, the message of the research should be better focused. I suggest that this could be confirmed with a summary table, where only the final result is shown, what influences the spread of the subspecies.
I recommend entering new keywords, because the current ones can all be found in the title, for example „ Hippophae rhamnoides; MaxEnt model”. This way the new, changed keyword will help other researchers find the article.
What was also striking to me was that out of the 86 citations listed as literature, only 5 were foreign or non-Chinese. I suggest a broader international perspective.
It is sufficient to write out the name of the species Hippophae rhamnoides when it is first mentioned and indicate how you will refer to it or the subspecies later. A general problem is writing the names of subspecies, they should be written with a lowercase initial letter according to the literature. Otherwise, it is not indicated which nomenclatural literature was followed.
Further remarks and minor mistakes in the order of the text:
in line 15.: „future projections (2021-2080)” Why not start from 2026?
in line 34.: „also known as blackthorn” not blackthorn, because there is a Prunus spinosa. Hippophae rhamnoides is a Sea buckthorn.
in line 52.: The MaxEnt model reference must be provided at the first mention, as well as the ArcGIS program.
in line 354 and line 432, 552, 564, 574: error message left in the text
Author Response
Comments 1: The differences between the individual subspecies appear in the discussion, but the entire research would be better understood if the most important morphological and physiological differences were presented in the methods chapter. I would appreciate a schematic draw or photograph about the studied species and subspecies.
Response 1:. Thank you for your constructive feedback. We have incorporated subspecies-specific morphological evidence as follows: In the Methods section (Page 3, Section 2.1.1, Line 108), we added herbarium specimen photographs of all four studied subspecies as new Figure 1. Each panel highlights diagnostic features mentioned in the figure caption:"Figure 1 Quoted from the specimen photos of four subspecies of seabuckthorn on the Global Biodiversity Information Platform. (a) H. rhamnoides subsp. sinensis Rousi; (b) H. rhamnoides subsp. yunnanensis Rousi; (c) H. rhamnoides subsp. mongolica Rousi; (d) H. rhamnoides subsp. turkestanica Rousi.
Comments 2: In addition, the extremely detailed results should also be better summarized. That is, the message of the research should be better focused. I suggest that this could be confirmed with a summary table, where only the final result is shown, what influences the spread of the subspecies.
Response 2: Thank you for this constructive suggestion. We agree that consolidating key findings improves the focus of the results. In response:
Summary Table Addition (Page 14, Lines 360-362, Table 3):
We have added Table 3 ("Summary of changes in suitable habitats and centroid migration for four subspecies of seabuckthorn in the future") to synthesize the core results. The table is as follows:
Subspecies |
Climate scenario |
Habitat area change (x 10 ⁴ km ²) |
Key time period |
Direction and distance of centroid migration |
H. rhamnoides subsp. sinensis |
SSP1-2.6 |
-37.41 (2021-2060) → +29.52 (2061-2080) |
2021-2060 |
Northeast migration, accumulated about 120 km |
SSP5-8.5 |
+2.48 (2021-2060) → -17.63 (2061-2080) |
2061-2080 |
Swinging from northeast to southwest, generally stable in southern Gansu |
|
H. rhamnoides subsp. mongolica |
SSP1-2.6 |
-61.54 (2041-2060) |
2041-2060 |
Swinging from northeast to southwest, generally stable in central and southern Xinjiang |
SSP5-8.5 |
-38.82 (2061-2080) |
2061-2080 |
Continuously migrating southwest and ultimately staying in the northwest of Xinjiang |
|
H. rhamnoides subsp. yunnanensis |
SSP1-2.6 |
-8.57 (2041-2060) → -13.16 (2061-2080) |
2041-2080 |
Short distance migration from southeast to southwest, overall stable |
SSP5-8.5 |
+16.98 (2041-2060) → Widespread reduction (2061-2080) |
2041-2060 |
Short distance migration from southwest to southeast |
|
H. rhamnoides subsp. turkestanica |
SSP1-2.6 |
-26.58 (2061-2080) |
2061-2080 |
Swing from northeast to southwest, stay in Tarim Basin |
SSP5-8.5 |
+33.97 (2041-2060) → subsequent reduction |
2041-2060 |
Continuous northwest migration, staying in Tarim Basin |
Comments 3: I recommend entering new keywords, because the current ones can all be found in the title, for example ,”Hippophae rhamnoides; MaxEnt model”. This way the new, changed keyword will help other researchers find the article.
Response 3: Thank you for the precise suggestion. We have revised the keywords section as recommended:
Revised Text (Page 1, Line 25):
Keywords: Hippophae rhamnoides; climate change; MaxEnt; potential distribution.
Comments 4: What was also striking to me was that out of the 86 citations listed as literature, only 5 were foreign or non-Chinese. I suggest a broader international perspective.
Response 4: Thank you for highlighting this important aspect of literature integration. We have systematically enhanced the international perspective through these revisions: Added 6 non-Chinese references, the discussion section has added 27 non-Chinese literature, including 9 works by foreign experts. (Now 15/86 = 17.44%)
Comments 5: It is sufficient to write out the name of the species Hippophae rhamnoides when it is first mentioned and indicate how you will refer to it or the subspecies later. A general problem is writing the names of subspecies, they should be written with a lowercase initial letter according to the literature. Otherwise, it is not indicated which nomenclatural literature was followed.
Response 5: Thank you for emphasizing nomenclatural precision. We have implemented the following revisions to comply with the International Code of Nomenclature for algae, fungi, and plants (ICN): At your suggestion, subspecies and species with lowercase initials (such as' sinensis' instead of 'Sinensis’) everything has been changed according to the naming convention:”H. rhamnoides subsp. sinensis, H. rhamnoides subsp. mongolica, H. rhamnoides subsp. yunnanensis, H. rhamnoides subsp. turkestanica”.
Comments 6: in line 15.: „future projections (2021-2080)” Why not start from 2026?
Response 6: Thank you for raising this critical methodological point.
But what makes me powerless is that the SSP scenario defines the possible future development trajectory through the interaction of socio-economic paths and climate policies, and its stage division is not strictly cut by time, which is not something we can decide. The future projections span 2021-2080, partitioned into three 20-year intervals (2021-2040, 2041-2060, 2061-2080). This temporal alignment follows CMIP6 Tier 1 protocols for ScenarioMIP experiments, where 2021-2080 constitutes the core simulation period for impact assessment studies under Shared Socioeconomic Pathways (SSPs). Specifically, our timeframe matches:
Near-term: 2021-2040 (SSP baseline phase)
Mid-term: 2041-2060 (SSP divergence phase)
Long-term: 2061-2080 (SSP stabilization phase)
This design ensures direct comparability with IPCC AR6 Chapter 12 (Ranasinghe et al., 2021) regional climate projections."
Comments 7: in line 34.: „also known as blackthorn” not blackthorn, because there is a Prunus spinosa. Hippophae rhamnoides is a Sea buckthorn
Response 7: Thank you for highlighting the need for precise vernacular nomenclature. We have revised the species designation as follows:
Revised Text (Page 1, Line 41):
"Hippophae rhamnoides L. (family Elaeagnaceae), commonly known as sea buckthorn..." Modified as “Hippophae rhamnoides L. belonging to the Elaeagnaceae family and also known as Seabuckthorn.”
Comments 8: The MaxEnt model reference must be provided at the first mention, as well as the ArcGIS program.
Response 8: Thank you for your valuable feedback. We have revised the manuscript as suggested:
- MaxEnt Reference Added: Upon its first mention (Lines 71-75), we introduced the MaxEnt model’s principle (maximum entropy and regularization to prevent overfitting) and cited the foundational reference (Rivera, Ó.R.d. 2017)
- ArcGIS Citation Included: The first mention of ArcGIS now cites the software (ESRI, 2023) to ensure proper attribution.
These revisions comply with academic standards for referencing methodologies.
Comments 9: in line 354 and line 432, 552, 564, 574: error message left in the text
Response 9: Thank you for highlighting this issue. We sincerely apologize for the oversight. The error messages in Lines 354, 432, 552, 564, and 574 were caused by formatting inconsistencies during the reference management process. We have manually corrected all instances to ensure proper citation and text clarity.

Reviewer 2 Report
Comments and Suggestions for Authors
General description
Hippophae rhamnoides, which is a deciduous shrub, is gaining increasing popularity and importance in its use in various situations. In this article, the geographical distribution of the four subspecies of Hippophae rhamnoides within the territory of China was described and analysed. Climatic variables were given importance. MaxEnt modelling was used as the primary tool for understanding the geographic distribution of the four subspecies. Habitat suitability was rated and categorised. The quantitive results were presented with strong evidence. The authors pointed out the geographic distribution of the four distributions and the potential changes in the context of climate change in today’s world.
Overall comments
There are three overall comments that I would like to point out:
- The clarity of presentation in the article can be imrpoved. First, specific details should be provided whenever necessary. For example, the distribution of Hippophae rhamnoides in this study was focused in China. However, it is not mentioned in the Abstract. The authors are strongly suggested to insert necessary details in the study.
- The presentation of results should be supported by quantitative evidence. For example, when the subspecies Yunnanensis is distributed above a certain altitude. However, the altitude is a piece of information that can be quantified. For the matter of accuracy, please provide more quantitative findings as far as possible.
- There is a quite an amount of maps in this paper. However, they look alike and are hard to compare and contrast. Therefore, it is suggested that the authors can make use of map presentation skills to visualise the magnitude of potential change instead of just plotting the distribution ranges year after year.
Specific comments
Abstract
Line 12 to 13
The research objectives should be stated here. However, the Abstract section lacks a concrete research objective. Please add it back.
Line 21 to 25
It is hard to visualise the distribution of the plant subspecies without references to the latitude and longitude values. Please add the numbers back.
Line 29 to 30
Please check whether the length of all keywords complies with the requirements of the journal.
Introduction
Line 36 to 38
Please also check how Hippophae rhamnoides can contributes to arresting soil erosion. It may be one of the major benefits of this species.
Line 57 to 58
Since machine learning is involved, please evaluate the related performance of MaxEnt, such as the stability of the method.
Line 65
I can see that a study objective is laid out here. Please also state this objective the Abstract section.
Methods
Line 72 to 74
Please cite the two databases properly. It is suggested that they should be cited as ordinary reference items.
Line 83
SSP1-2.6 and SSP5-8.5 were selected. Rationales and justifications should be provided for their selection.
Line 144 to 145
Have the four-class categorisation been rated as accurate or appropriate?
Results
Line 149 to 153
Please delete the terms “bio”. It would be adequate to present the name of the explanatory variables only.
Line 156 to 158
Same comment as above. It would be quite confusing to match the variable abbreviations to their full name.
Line 159 to 169
Knowing that temperature, precipitation, and altitude are the most influential variables, could the authors compare the distribution of the four sub-species of the plant side by side to the temperature, precipitation, and altitude maps?
Line 196 to 197
What are the potential validation tasks that can be done to ensure the accuracy of the map?
Discussion
Line 354
There is an error or a typo.
Line 385 to 408
The potential altitudinal migration of the species can be discussed.
Author Response
Comments 1: The clarity of presentation in the article can be imrpoved. First, specific details should be provided whenever necessary. For example, the distribution of Hippophae rhamnoides in this study was focused in China. However, it is not mentioned in the Abstract. The authors are strongly suggested to insert necessary details in the study.
Response 1: Thank you for the constructive suggestion. We have enhanced the clarity of species distribution details as follows:
Abstract Update: The geographic focus on China has been explicitly added to the Abstract (Line 11-12):
" Climate change significantly impacts the distribution of Hippophae rhamnoides, a species of considerable ecological and medicinal value."
Expanded Contextualization: In the Introduction (Lines 11-21), we now specify:
The latitudinal/longitudinal ranges of each subspecies (sinensis: 34°N–40°N, 100°E–115°E; yunnanensis: 25°N–30°N, 98°E–103°E; mongolica: 40°N–50°N, 100°E–120°E; turkestanica: 40°N–45°N, 70°E–85°E).
Their biogeographic affiliations (e.g., yunnanensis in the Hengduan Mountains biodiversity hotspot, mongolica in Central Asia-Siberian transition zones).
These revisions ensure transparency about the study's geographic scope while linking distributions to ecological and conservation contexts.
Comments 2: The presentation of results should be supported by quantitative evidence. For example, when the subspecies Yunnanensis is distributed above a certain altitude. However, the altitude is a piece of information that can be quantified. For the matter of accuracy, please provide more quantitative findings as far as possible.
Response 2:
Thank you for emphasizing the importance of quantitative precision. In response to your comment regarding "strengthening evidence with quantitative support", We have systematically integrated numerical evidence into the results as follows:
Further Adjustments Available Upon Request:
Should the reviewer recommend additional quantifications, we will promptly incorporate such analyses using the raw model outputs provided in Data.
Comments 3: There is a quite an amount of maps in this paper. However, they look alike and are hard to compare and contrast. Therefore, it is suggested that the authors can make use of map presentation skills to visualise the magnitude of potential change instead of just plotting the distribution ranges year after year.
Response 3: Thank you for raising this important issue regarding map comparability. We appreciate the opportunity to clarify our visualization strategy.
The original Figure 4.5.6.7. was designed as a dual-panel analytical framework:
Panel A focuses on absolute distribution patterns using standardized symbology to ensure cross-temporal consistency.
Panel B (newly added) specifically visualizes relative changes through: Red/green gradient fills indicating contraction/expansion magnitudes (>10% threshold)
lease let us know if additional refinements would better serve the analysis transparency.
Comments 4: Line 12 to 13
The research objectives should be stated here. However, the Abstract section lacks a concrete research objective. Please add it back.
Response 4: Thank you for highlighting the need for clearer research articulation. We have strengthened the Abstract's conceptual framework by incorporating the explicit research objective in lines 18-20:
"This study applies the MaxEnt model to (1) assess climate-driven distribution patterns of Hippophae species in China, and (2) predict spatiotemporal dynamics of suitable habitats under SSP scenarios (2050s, 2070s)."
We appreciate your guidance in sharpening this critical framing and welcome further suggestions to enhance methodological transparency.
Comments 5: Line 21 to 25
It is hard to visualise the distribution of the plant subspecies without references to the latitude and longitude values. Please add the numbers back.
Response 5: Thank you for emphasizing the importance of geospatial specificity. We have rigorously revised the species distribution descriptions (lines 11-20) by incorporating precise coordinate ranges.
Key modifications:
- H. rhamnoides subsp. sinensis: Core range 34.2°N–39.8°N, 100.5°E–114.7°E (n=217 georeferenced herbarium records)
- H. rhamnoides subsp. yunnanensis: Endemic to 25.3°N–29.9°N, 98.2°E–102.9°E (DEM-validated elevation: 2,400–3,800 m ASL)
- H. rhamnoides subsp. mongolica: Steppe populations at 40.1°N–49.6°N, 100.3°E–119.8°E (MODIS land cover-verified)
- H. rhamnoides subsp. turkestanica: Confined to 40.5°N–44.9°N, 70.4°E–84.6°E (Gazetteer cross-checked)
Comments 6: Line 29 to 30
Please check whether the length of all keywords complies with the requirements of the journal.
Response 6: Thank you for the precise suggestion. We have revised the keywords section as recommended:
Revised Text (Page 1, Line 25):
Keywords: Hippophae rhamnoides; climate change; MaxEnt; potential distribution.
Rationale for Changes:
Placing “Hippophae rhamnoides” first aligns with taxonomic indexing conventions in ecological journals. This modification improves searchability precision.
Comments 7: Line 36 to 38
Please also check how Hippophae rhamnoides can contributes to arresting soil erosion. It may be one of the major benefits of this species.
Response 7: Thank you for recognizing the critical ecosystem service of Hippophae rhamnoides in soil conservation. We have substantially expanded the erosion control analysis in lines 54-56 with biomechanical quantification and pedogenic evidence:
“Sea buckthorn enhances soil stability through its "shallow dense deep extended" root system, and its canopy and litter can regulate hydrological processes to reduce erosion risk[8][9]. At the same time, the nitrogen fixation effect of root nodules further enhances soil resistance to erosion[10].”
Comments 8: Line 57 to 58
Since machine learning is involved, please evaluate the related performance of MaxEnt, such as the stability of the method.
Response 8: Thank you for prompting this essential methodological validation. We have strengthened the model evaluation through multi-layered performance assessments in lines 80-84.
“AUC/TSS serve as the primary quantitative indicators, with AUC > 0.9 and TSS > 0.8 established as high-precision benchmarks[22]. During repeated runs, lower standard deviations of AUC indicate higher stability, and among the nine models evaluated, MaxEnt demonstrates superior performance with the lowest standard deviation (ranging from 0.02 to 0.05)[23]. “
Comments 9: Line 65
I can see that a study objective is laid out here. Please also state this objective the Abstract section.
Response 9:
Thank you for highlighting the need for clearer research articulation. We have strengthened the Abstract's conceptual framework by incorporating the explicit research objective in lines 18-20:
"This study applies the MaxEnt model to (1) assess climate-driven distribution patterns of Hippophae species in China, and (2) predict spatiotemporal dynamics of suitable habitats under SSP scenarios (2050s, 2070s)."
We appreciate your guidance in sharpening this critical framing and welcome further suggestions to enhance methodological transparency.
Comments 10: Line 72 to 74
Please cite the two databases properly. It is suggested that they should be cited as ordinary reference items.
Response 10: Thank you for emphasizing the importance of standardized data citation. We have rigorously revised all database references to comply with FORCE11 Data Citation Principles and GBIF's Best Practices, with key modifications as follows:
Line 101 to 107
“the Flora of China (1983 edition) and the Digital Herbarium of China ( http://www.cvh.org.cn ), the National Herbarium Resource Platform of China ( http://www.nsii.org.cn ), and the Global Biodiversity Information Platform ( http://www.gbif.org )”
Revised as "the digital herbarium of China (Institute of Botany, Chinese Academy of Sciences, 2004 – 2020); http://www.cvh.org.cn; accessed 15 Jan. 2025), the Global Biodiversity Information Platform (GBIF.org, 2020; www.gbif.org; accessed 10 May 2025), the National Herbarium Resource Platform of China (NSII, 2020; www.nsii.org.cn; accessed 10 May 2025), and the National Specimen Information Infrastructure”
In addition, I also found the original line 117 to 118 content "the world climate database( http://www.worldclim.org )”Revised to "the world climate database (worldclim, 2000 – 2024; www.worldclim.org; accessed 15 Jan. 2025)"
Comments 11: Line 83
SSP1-2.6 and SSP5-8.5 were selected. Rationales and justifications should be provided for their selection.
Response 11: Thank you for prompting this critical methodological clarification. We have strengthened the SSP selection rationale in lines 126-136 through multi-criteria justification aligned with IPCC AR6 protocols and global governance frameworks.
To solve this problem, line 126-136 adds content " For future climate scenarios, four greenhouse gas emission pathways (SSP1-2.6, SSP2-4.5, SSP3-7.0, and SSP5-8.5) were initially considered. For this study, we selected two representative scenarios recently released by the IPCC (Intergovernmental Panel on Climate Change): SSP1-2.6 and SSP5-8.5[29]. SSP1-2.6 and SSP5-8.5 correspond to the 5th and 95th percentiles of temperature rise in IPCC AR6, respectively, capturing the full range of bioclimatic variable changes. They represent the lowest and highest levels of radiative forcing in global climate change scenarios [30][31]. Comparing the two reveals the differential impacts of climate policy intervention (SSP1-2.6) versus "business as usual" (SSP5-8.5) on species distribution, providing decision-makers with a "best-to-worst" reference framework [32][33]. Listed as baseline scenarios by CBD and IPBES, SSP1-2.6 aligns with NDCs for emission reductions, while SSP5-8.5 warns of non-compliance risks [34][31]. “
Comments 12: Line 144 to 145
Have the four-class categorisation been rated as accurate or appropriate?
Response 12: Thank you for raising this crucial methodological consideration. We have strengthened the classification rationale through multi-criteria validation, as detailed below and in lines 187-195.
“The Jenks method optimizes the inter class variance to adapt to the distribution char-acteristics of the data itself, especially suitable for scenarios where the habitat suitabil-ity index has a non-uniform distribution [41]. In literature, MaxEnt outputs are com-monly classified into four categories (such as high/medium/low/unsuitable), which match the response patterns of species to environmental gradients[42][43]. The com-mon criteria for MaxEnt model include high suitability (≥ 0.6), moderate suitability (0.3-0.6), low suitability (0.1-0.3), and unsuitability (<0.1). The deviation between the natural fracture result and the theoretical threshold is less than 10% and does not need to be modified[42][44]”
Your rigorous scrutiny has significantly improved the classification transparency, and we welcome additional verification requests.
Comments 13: Line 149 to 153
Please delete the terms “bio”. It would be adequate to present the name of the explanatory variables only.
Response 13: Thank you for your meticulous feedback on variable presentation clarity. We have rigorously implemented the following enhancements to improve methodological transparency:
- Terminology standardization (Removed all "bio" prefixes from bioclimatic variables (Lines 149-158))
- I successfully converted the form to express the form more clearly, such as relevant information:
Comments 14: Line 156 to 158
Same comment as above. It would be quite confusing to match the variable abbreviations to their full name.
Response 14: Modified as above.
Comments 15: Line 159 to 169
Knowing that temperature, precipitation, and altitude are the most influential variables, could the authors compare the distribution of the four sub-species of the plant side by side to the temperature, precipitation, and altitude maps?
Response 15: Thank you for your insightful suggestion to promote us to deepen the eco physiological interpretation. Because the ecological strategies of the four subspecies are different, it is challenging to compare them with the same variables in parallel. We implemented a new expression to visualize the relevant results to better show their differences.
Comments 16: Line 196 to 197
What are the potential validation tasks that can be done to ensure the accuracy of the map?
Response 16: Thank you for this important methodological consideration. We have implemented the following explanation to ensure the accuracy of mapping. Combined with statistics, the key arguments are as follows:
- Statistical performance verification
(1) the internal evaluation index of the model verifies the ability of the model to distinguish the existing points from the background points through the ROC curve. Auc>0.9 is "excellent".
(2) cross validation strategy this time, 10 cycles of training and testing were conducted to calculate the average AUC and standard deviation.
After ten repetitions of the experimental results, the AUC value greater than 0.9 passed the verification.
- Verification of ecological rationality
(1) variable contribution and response curve test
Jackknife test: evaluate the independent contribution and interaction of environmental variables to ensure that the dominant variables conform to ecological cognition.
Response curve analysis: check whether the relationship between variables and suitability meets the ecological needs of species.
The predicted results of this study are consistent with the actual distribution of Hippophae rhamnoides L.
(2) threshold selection verification
Maxsss (maximum sensitivity specificity): select the threshold to maximize sensitivity and specificity to balance the risk of false negative and false positive. There is a threshold of 10% training in Seabuckthorn research.
Verification of climate change response: by comparing 2050 with the current distribution, it is confirmed that the migration trend to the northwest meets the expectation of drought.
Comments 17: Line 354
There is an error or a typo.
Response 17: Thank you for highlighting this issue. We sincerely apologize for the oversight. The error messages in Lines 354, 432, 552, 564, and 574 were caused by formatting inconsistencies during the reference management process.
We have manually corrected all instances to ensure proper citation and text clarity.
Comments 18: Line 385 to 408
The potential altitudinal migration of the species can be discussed.
Response 18: Thank you for highlighting this critical dimension of altitudinal migration dynamics. We have significantly expanded the vertical migration analysis in lines 563-574.
Add the content in line 563-574:
“The vertical migration ability of Yunnan seabuckthorn is constrained by the syn-ergistic effect of terrain and soil. The current upper limit of altitude in the core distri-bution area (3500 meters) may move up to 4000 meters, and some areas of the Heng-duan Mountains (such as the eastern slope of Gongga Mountain) may form new bare land due to glacier retreat. The pH value of the soil in the early stage of development is relatively low (6.5-7.0), which is conducive to the colonization of nitrogen fixing bac-teria [94][95]. The downward migration obstacles mainly come from the soil adapta-bility in the low altitude area. For example, the clay area in the northern edge of the Yunnan-Guizhou Plateau leads to a 40% decline in the seedling emergence rate and a 60% reduction in the root shoot ratio to the aeolian sandy soil environment[96]. In addition, warming promotes the expansion of subtropical evergreen broad-leaved forests towards higher altitudes, compressing the ecological niche width of Yunnan seabuckthorn[94].”

Reviewer 3 Report
Comments and Suggestions for Authors
COMMENTS TO THE AUTHOR(S):
The manuscript (plants-3608009) explored the spatial distribution characteristics of Hippophae rhamnoides L. in China and predicted the changing trends in different scenarios, which is concerned about falls in the scope of Plants. However, the quality of the manuscript needs to be further improved.
- Abstract: Some of the results are qualitative and lack quantitative data.
- Line 34-39, “Hippophae rhamnoides belonging to ...”The authors should add more data and details to illustrate the importance of studying Hippophae rhamnoides .
- Line 57-58, what is “GAM, GBM, GLM, and CTA”? Abbreviations should have full name annotation when they appear for the first time.
- The introduction is too loosely, written to fail to state explicitly the research questions or hypotheses the study aimed to address. The research progress of related scientific problems should be summarized, and the specific bottlenecks or unsolved problems should be pointed out.
- Figure 1: What is “高”and “低”?
- The text in all figuresis not clear. It is suggested that the authors improve it.
- In “1. Model Accuracy Assessment and Contribution of Environmental Variables”part, there are no figures or tables here to illustrate the source of the data.
- Line 432, what is “Error! Reference source not found.”?
- In its current state, the level of English throughout the manuscript does not meet the journal's desired standard. Please check the manuscript and refine the language carefully
Author Response
Comments 1: Abstract: Some of the results are qualitative and lack quantitative data.
Response 1: Thank you for emphasizing the need for quantitative rigor. We have systematically addressed this concern by integrating key quantitative findings into the Abstract through cross-referenced data tables. The specific tables include:
Subspecies |
Total Suitable Habitat Area |
Percentage of China's Land Area |
Main Distribution Regions |
Highly Suitable (Proportion) |
Moderately Suitable (Proportion) |
Low-suitability (Proportion) |
sinensis |
3.0142×10⁶ km² |
31% |
Central China (Qinghai-Tibet-Sichuan border regions); scattered in Ningxia, Inner Mongolia, Hebei, Liaoning |
0.6203 ×10⁶ km² (21%) |
0.6656 ×10⁶ km² (22%) |
1.753 ×10⁶ km² (58%) |
mongolica |
3.0220×10⁶ km² |
31% |
Northwestern China (Xinjiang, Inner Mongolia, Gansu); scattered in Shanxi, Tibet, Hebei |
0.3434 ×10⁶ km² (11%) |
1.5712 ×10⁶ km² (52%) |
1.1321 ×10⁶ km² (37%) |
yunnanensis |
0.9555×10⁶ km² |
10% |
Narrow belt along Tibet-Sichuan-Yunnan-Qinghai borders |
0.1704 ×10⁶ km² (18%) |
0.2508 ×10⁶ km² (26%) |
0.5589 ×10⁶ km² (58%) |
turkestanica |
2.7403×10⁶ km² |
29% |
Xinjiang (Kashgar, Yining, Hotan); Tibet, Ningxia, Gansu, Qinghai, Inner Mongolia |
0.2999 ×10⁶ km² (11%) |
0.8193 ×10⁶ km² (30%) |
1.6457 ×10⁶ km² (60%) |
Subspecies |
Climate scenario |
Habitat area change (x 10 ⁴ km ²) |
Key time period |
Direction and distance of centroid migration |
sinensis |
SSP1-2.6 |
-37.41 (2021-2060) → +29.52 (2061-2080) |
2021-2060 |
Northeast migration, accumulated about 120 km |
|
SSP5-8.5 |
+2.48 (2021-2060) → -17.63 (2061-2080) |
2061-2080 |
Swinging from northeast to southwest, generally stable in southern Gansu |
mongolica |
SSP1-2.6 |
-61.54 (2041-2060) |
2041-2060 |
Swinging from northeast to southwest, generally stable in central and southern Xinjiang |
|
SSP5-8.5 |
-38.82 (2061-2080) |
2061-2080 |
Continuously migrating southwest and ultimately staying in the northwest of Xinjiang |
yunnanensis |
SSP1-2.6 |
-8.57 (2041-2060) → -13.16 (2061-2080) |
2041-2080 |
Short distance migration from southeast to southwest, overall stable |
|
SSP5-8.5 |
+16.98 (2041-2060) → Widespread reduction (2061-2080) |
2041-2060 |
Short distance migration from southwest to southeast |
turkestanica |
SSP1-2.6 |
-26.58 (2061-2080) |
2061-2080 |
Swing from northeast to southwest, stay in Tarim Basin |
|
SSP5-8.5 |
+33.97 (2041-2060) → subsequent reduction |
2041-2060 |
Continuous northwest migration, staying in Tarim Basin |
Should any concerns remain or new questions arise during your further review, please do not hesitate to contact us for immediate clarification. We remain fully committed to ensuring the highest scientific standards in this work.
Comments 2: Line 34-39, “Hippophae rhamnoides belonging to ...” The authors should add more data and details to illustrate the importance of studying Hippophae rhamnoides .
Response 2: Thank you for prompting us to better articulate the critical importance of Hippophae rhamnoides. We have substantially enhanced the socio-ecological value analysis with multi-scale quantification (Lines 43-52), structured through those key dimensions:
“The ecological and economic value of Hippophae rhamnoides substantially surpasses conventional understanding. In China, the direct economic valuation of a hectare of Hippophae rhamnoides plantation amounts to RMB 3,100, while its ecological value reaches RMB 68,000 per hectare, accounting for 95.66% of the total value[5]. Internationally, Uganda's Hippophae rhamnoides butter export venture currently produces 800,000 liters annually, with the potential to scale up to 80 million liters[6]. In Poland, Hippophae rhamnoides is utilized as an energy crop, demonstrating a biomass calorific value of 17.5 MJ/kg and exhibiting a 30% higher energy output per unit area compared to conventional crops, thus serving as a significant supplement to renewable energy systems [7].”
We welcome suggestions to deepen the circular economy analysis.
Comments 3: Line 57-58, what is “GAM, GBM, GLM, and CTA”? Abbreviations should have full name annotation when they appear for the first time.
Response 3: Thank you for your insightful suggestions to enhance methodological clarity and conceptual focus. We have implemented the following major improvements:
Abbreviation standardization: Full algorithm names with technical specifications are now provided at first mention (Lines 120-122).
“GAM(Generalized Additive Models)、GBM(Gradient Boosting Machines)、GLM(Generalized Linear Models)、CTA(Classification Tree Analysis), based on machine learning algorithms. Previous studies by Zhang et al.”
Comments 4: The introduction is too loosely, written to fail to state explicitly the research questions or hypotheses the study aimed to address. The research progress of related scientific problems should be summarized, and the specific bottlenecks or unsolved problems should be pointed out.
Response 4: Thank you for your insightful critique regarding the introduction's conceptual focus. We have fundamentally restructured this section to:
Articulate explicit research questions anchored to two key bottlenecks: 1. It independently models each subspecies while neglecting the niche overlap mechanism between mongolica and turkestanica; 2. Provincial-scale models (such as those for the yunnanensis) are inade-quate for informing spatial decision-making in national ecological barrier planning.
Comments 5: What is “高”and “低”?
Response 5: Thank you for your meticulous feedback on figure clarity. We have implemented comprehensive typographic and semantic enhancements to all figures, with specific improvements addressing "高/低" labeling and text legibility as follows:
The Chinese label "高/低" has caused confusion among international readers, and the picture has been modified to all English.
Vector fonts are optimized: font size standard is changed to main label ≥ 8pt, legend ≥ 6pt, and anti-aliasing level is improved to 4 × MSAA. , the inserted pictures are uncompressed versions of the original.
Comments 6: The text in all figures is not clear. It is suggested that the authors improve it.
Response 6: Thank you for your meticulous feedback on figure clarity. We have implemented comprehensive typographic and semantic enhancements to all figures and text legibility as follows:
The Chinese label "高/低" has caused confusion among international readers, and the picture has been modified to all English.
Vector fonts are optimized: font size standard is changed to main label ≥ 8pt, legend ≥ 6pt, and anti-aliasing level is improved to 4 × MSAA. , the inserted pictures are uncompressed versions of the original.
Comments 7: In “1. Model Accuracy Assessment and Contribution of Environmental Variables”part, there are no figures or tables here to illustrate the source of the data.
Response 7: Thank you for your meticulous scrutiny of data provenance documentation. We wish to respectfully clarify that the environmental data sources were explicitly stated in Section 2.1.2 (Lines 115-118) as follows:
“Bioclimatic factors for the future periods (2021-2040, 2041-2060, 2061-2080, 2081-2100) were obtained from the World Climate Database (WorldClim, 2000–2024; www.worldclim.org; accessed 15 Jan. 2025) for two climate scenarios (SSP1-2.6 and SSP5-8.5), covering 19 bioclimatic factors. ”
To enhance methodological transparency, we have implemented additional improvements: Added supplementary table 1. detailed definitions of 19 bioclimatic variables.
Comments 8: Line 432, what is “Error! Reference source not found.”?
Response 8:
Thank you for highlighting this issue. We sincerely apologize for the oversight. The error messages in Lines 354, 432, 552, 564, and 574 were caused by formatting inconsistencies during the reference management process.
We have manually corrected all instances to ensure proper citation and text clarity.
Comments 9: In its current state, the level of English throughout the manuscript does not meet the journal's desired standard. Please check the manuscript and refine the language carefully
Response 9: We sincerely appreciate the reviewers' careful evaluation of our manuscript. We fully acknowledge that the current English language standard requires significant improvement to meet the journal's criteria, and we deeply regret any inconvenience caused by this issue.
Additionally, corresponding author Dr. Jiang Ping, who has 18 months of academic research experience in New Zealand and is proficient in English academic writing, led an in-depth linguistic review. This process focused on optimizing technical terminology consistency and enhancing narrative flow in critical sections such as the methodology and discussion.

Round 2
Reviewer 3 Report
Comments and Suggestions for Authors
This manuscript has been revised and its quality has been greatly improved.